# Differential relationship between meditation methods and psychotic-like and mystical experiences

Timothy Palmer[1,2]*, Kenza Kadri[1,2], Eric Fakra[3,4], Jacqueline Scholl[5,6°], Elsa Fouragnan[1,2°]

1 Faculty of Health, School of Psychology, University of Plymouth, Plymouth, United Kingdom, 2 Brain Research Imaging Center, Faculty of Health, University of Plymouth, Plymouth, United Kingdom, 3 University Jean Monnet Saint-Etienne, CHU Saint-Etienne, Saint-Etienne, France, 4 PSYR2, CNRL, INSERM U1028, CNRS UMR5292, UCBL1, Bron, France, 5 INSERM, CRNL U1028 UMR5292, PsyR2, Centre Hospitalier le Vinatier, Université Claude Bernard Lyon 1, BRON Cedex, France, 6 Oxford Centre of Human Brain Activity, Wellcome Integrative Neuroimaging (WIN), Department of Psychiatry, University of Oxford, Oxford, United Kingdom

☯ These authors contributed equally to this work.
* timothy.palmer@plymouth.ac.uk

**Data Availability Statement:** The data and scripts for analysis are available via the OSF: https://osf.io/gxwcn/.

## Abstract

Much work has investigated beneficial effects of mindfulness-based meditation methods, but less work has investigated potential risks and differences across meditation methods. We addressed this in a large pre-registered online survey including 613 mediators where we correlated participants' experience with fifty meditation techniques to psychotic-like experiences (PLEs) and mystical experiences. We found a positive correlation for both PLEs and mystical experiences with techniques aiming at reducing phenomenological content ('null-directed', NDM) or classified as non-dual or less embodied. In contrast, methods aiming at achieving an enhanced cognitive state (CDM), also described as 'attentional' or strongly embodied, showed negative correlations with PLEs. Interestingly, participants' subjectively perceived that all types of meditation techniques were preventative of PLEs but less so for NDM. Participants differed in their reasons for meditating, broadly grouped into associated with spiritual exploration and associated with health. Participants who meditated for spiritual reasons were more likely to choose NDM techniques and more likely to experience PLEs. In contrast, participants who meditated for health-related reasons were more likely to choose CDM techniques. This study provides important information for meditators about the relationship of different techniques with PLEs and the moderating influences of individual traits.

## Introduction

Many people start a meditation practice to reduce their stress and feel better [1]. A large body of work has highlighted the wide-ranging benefits meditation can bring to an individual [2, 3]. However, criticisms have been raised that potential negative side-effects have been overlooked [4–6]. Although a few studies have found that meditation can be associated with unpleasant

**Funding:** This study was financially supported by UK Research and Innovation (UKRI) Medical Research Council (MRC) (https://www.ukri.org/councils/mrc/) in the form of a Skills Development Fellowship award (MR/N014448/1) received by JS. This study was also financially supported by UK Research and Innovation (UKRI) Biotechnology and Biological Sciences Research Council (BBSRC) (https://www.ukri.org/councils/bbsrc/) in the form of a Discover Fellowship award (BB/V004999/1) received by JS, and an award (BB/Y001494/1) received by E. Fouragnan. This study was also financially supported by UK Research and Innovation (UKRI) in the form of a Future Leaders Fellowship award (MR/T023007/1) received by E. Fouragnan. This study was also financially supported by the Psychology Department, Faculty of Health, University of Plymouth (https://www.plymouth.ac.uk/about-us/university-structure/faculties/health) in the form of awards for TP and KK. This study was also financially supported by University Jean Monnet Saint-Etienne (https://www.univ-st-etienne.fr/en/index.html) in the form of awards for JS and E. Fakra. The funders had no role in study design, data collection and analysis, decision to publish, or preparation of the manuscript.

**Competing interests:** The authors have declared that no competing interests exist.

and adverse experiences [7, 8], and clinical cases have suggested that psychotic-like experiences (PLEs) could be triggered by meditation [9, 10], research looking at the negative consequences of mediation remains marginal. In our pre-registered cross-sectional survey study, we examined the links between different types of meditation and adverse (PLEs) and beneficial (mystical experiences) outcomes of meditation in a sample of 613 meditators.

## Psychotic-like experiences

Psychotic-like experiences (PLEs) are considered part of a continuum of psychosis. While people reporting such experiences do not always report distress or dysfunction [11] and their prevalence is high (>25% of the general population have had at least one PLE [12]), they are also associated with an increased risk of psychosis [13, 14]. PLEs can be measured by screening questionnaires such as the Community Assessment of Psychic Experiences-42 (CAPE-42) [15]. They consist of positive symptoms presented as hallucinations (visual and auditory), delusions and thought disorder, and negative symptoms including anhedonia, withdrawal from social interaction and depressive symptoms such as rumination [16]. Here we focus on positive symptoms.

## Meditation techniques and classification systems

Much research has examined Mindfulness Based Stress Reduction (MBSR) and Mindfulness Based Cognitive Therapy (MBCT) and their beneficial impact upon mental health [17]. MBCT is prescribed to run for 8 consecutive weeks and utilizes both formal and informal meditation practice. The formal practice consists of guided meditation sessions including walking meditations, body scan meditation, mindful movement (drawn from Hatha yoga) and encouragement to become more aware of daily routines (more 'mindful'). The cognitive therapy aspect includes learning about depression and how to deal with distressing thoughts, and setting action plans to identify symptoms and how to deal with them when they occur [18]. MBSR is also prescribed for 8 weeks with 2.5 hours of daily group work. MBSR also uses guided meditation, including body scan and yoga derived techniques focussed on bodily movement and sensations, and sitting meditations focussed on developing a non-judgemental an accepting manner towards thoughts and emotions [19]. These meditation interventions can be summarised as aiming at enhancing awareness of the present moment, a greater focus upon bodily signals, and a reduction in mind wandering [20]. We present an approach to utilising meditation taxonomies to better understand how the goal of different meditation techniques, the amount of focus upon bodily signals and potential cognitive mechanisms relate to PLEs, mystical experiences and traits linked to improved mental health.

Classifying meditation has proved an area of debate within meditation research, and several taxonomies for classification now exist [21]. An initial barrier to classifying meditation is to first identify the variety of techniques that exist across traditions. While some classifications categorize meditation techniques by their ultimate goals (e.g. Nash, Newberg and Awasthi's) [22], such as aiming to achieve a state of 'emptiness' or heightened focus upon an object, others classify them by cognitive mechanisms (e.g. Dahl et al) [23] and yet another approach has been to describe the techniques meditators engage in (e.g. Matko et al) [24, 25]. The latter approach has identified that there are many more (50) types of commonly used meditation techniques [24] (S1 in S1 File). One way these techniques have been parsed is along an axis of 'bodily focus' [24]. Techniques with a more abstract or conceptual focus (e.g., 'concentrating on a contradiction or paradox') lay at the low body-orientation end of the first dimension, with techniques such as 'scanning the body' and 'manipulating the breath' at the opposite high body-orientation end of the dimension. The second 'activation' dimension refers to the amount of

bodily activation involved with a technique. Techniques such as 'walking and observing senses' and 'manipulating the breath' are at the high activation end, and 'lying meditation' and 'observing thoughts or emotions' are at the low end [24].

In contrast, classifying meditation techniques according to goals has led to different, partly overlapping, frameworks (e.g. Dahl et al [23] and Nash et al [22]). According to Nash, Newberg and Awasthi's [22] taxonomy, utilizing a specific technique will lead towards an 'enhanced meditative state', whose contents is determined by the specific aim. Their three categories include Null Directed Methods (NDM), Affective Directed Methods (ADM), Cognitive Directed Methods (CDM). NDM techniques aim at an 'enhanced non-cognitive non-affective state' which is described as devoid of phenomenological contents, which in essence might be described as empty. An example of this comes from the Hindu tradition and is known as 'self-inquiry' and uses the question 'who am I' to encourage the meditator to become aware of thoughts, feelings and sensations as 'objects' in consciousness rather than who they truly are [25]. ADM aims at attaining an 'enhanced affective state' that consists of a heightened affective or emotional state. An example of an ADM technique is Tibetan Non-Referential Unconditional Loving-Kindness method which encourages a focus upon extending loving-kindness to all sentient beings [22]. CDM techniques, which in Nash et al's framework covers any techniques not labelled NDM or ADM, is described at aiming to achieve an 'enhanced cognitive state'. This consists of an enhanced focus upon a cognitive element, such as the focus upon an object in awareness like the breath, a sensation, or an aim such as to be more present focussed. Examples include mindfulness meditation (vipassana) and concentration meditation [22].

Dahl et al. [23] have identified cognitive mechanisms as key to categorising meditation techniques. They focus on mechanisms including meta-awareness (awareness of awareness), perspective taking and cognitive reappraisal and self-enquiry in relation to the outcomes of using different techniques. They divide techniques into three families, the 'Attentional Family', the 'Constructive Family' and the 'Deconstructive Family'. Attentional Family techniques are aimed at enhancing attentional regulation, including re-orienting attention, disengaging attention, and sustaining attention. Constructive Family techniques seek to alter maladaptive self-schema by replacing them with more adaptive concepts, which might include a focus on becoming more aware of what is meaningful to the individual for example. Deconstructive Family techniques are focussed on self-enquiry, described as the exploration of self-related processes [23]. These three categories are each further divided into three, with the Attentional Family consisting of 'Focussed Attention (FA), 'Open Monitoring Object-Orientation' (OMO) and 'Open Monitoring Subject-Orientation' (OM-S). The Constructive Family is divided into 'Relationship Orientation' (C-R), 'Values Orientation' (C-V) and 'Perception Orientation' (C-P). The Deconstructive Family has 'Object-Oriented Insight' (OO-I), 'Subject-Oriented Insight' (SO-I) and 'Nondual-Oriented Insight (ND) [23].

Dah et al's and Nash et al's taxonomies are argued to share some overlap and be complimentary to one another. In particular, Nash et al's three category approach presents a simple and straightforward way to categorise techniques. Dah et al's taxonomy provides further nuance, whereby techniques in each category are further divided by three and can be described in terms of differences in their key cognitive mechanisms. This approach is suggested useful in guiding hypothesis formation and the design of studies exploring the experiences associated with the use of different techniques. Matko et al's taxonomy further provides a way to categorise these techniques in relation to the amount of bodily focus they have, which is useful in terms of exploring related experiences of alterations to the bodily self (e.g. body sensations, bodily boundaries, and agency) that have been reported by meditators in relation to meditation practice [26]. Our approach primarily applied Nash et al's [22] taxonomy in the design of a survey study (see Fig 1A), with a combination of this taxonomy, Matko et al's and Dahl et al's

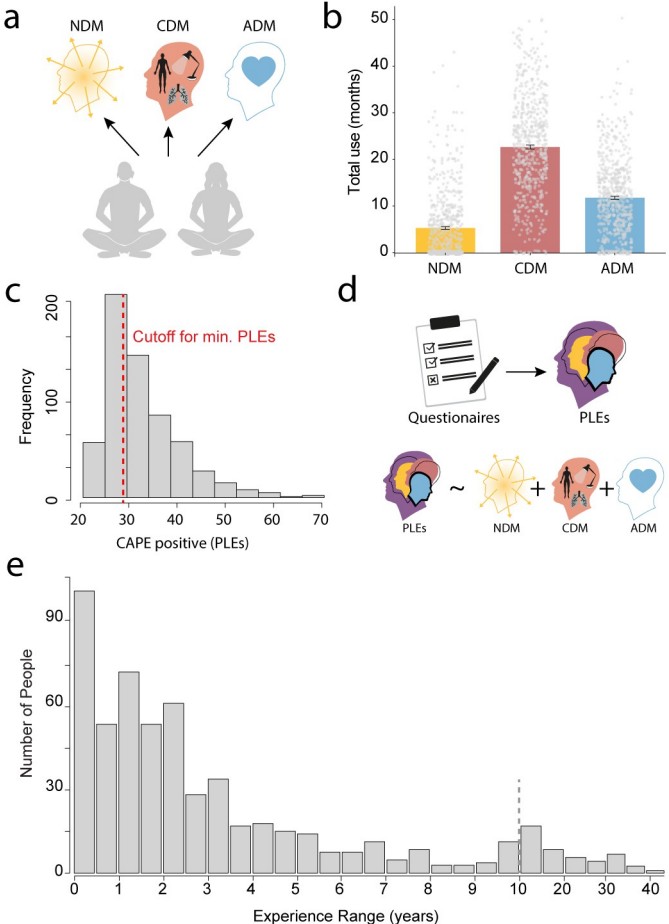

**Fig 1.** A. Meditation methods according to Nash, Newberg and Awasthi's taxonomy [22] (NDM = Null Directed Methods, CDM = Cognitive Directed Methods, ADM = Affective Directed Methods). B. Total lifetime use for each meditation method across participants from ratings 0 (Not at all) to 10 (Very often) for each of the five in each category and summed (i.e. minimum is 0 and maximum is 50). A participant could score highly in only one category or in several or in none. C. Distribution of PLE (CAPE positive) scores with a cut-off (red dotted line) for clinical significance. D. Pre-registered analyses: from questionnaires, psychotic-like experiences (PLEs) [15], and mystical experiences [38], were extracted to test how they were differentially affected by different types of meditation. E. Spread of overall lifetime meditation practice in years, with the dotted vertical line marking 10+ years experience.

in the analysis of the results (see Table 2). Multiple case series and a qualitative study have associated PLEs [9, 27–29] with meditation. These prior studies do not clearly delineate the specific categories of meditation associated with PLEs. Our study provides a quantitative test that applies meditation taxonomies to the study of PLEs and meditation.

## Reasons to meditate and personal preferences

Previous work has identified individual differences in the meditation techniques people prefer [30]. The reasons people start meditating, and choose to continue, may also be an important factor regarding the techniques they choose [1]. It is not clear whether the goals people have regarding intended outcomes from meditation have any influence upon the experiences they report, and if this is independent of the techniques they choose. How motivations to meditate evolve over time is also an interesting topic and might be informative as to why some studies

have found meditation experience to link to the likelihood of reporting an adverse experience [8]. Sedlmeier and Theumer [1] found that as meditation experience increases, that motivations tend to shift towards seeking spiritual experiences and insights. They also suggest that whether the meditator reports following a tradition also influences their motivations, with those reporting a spiritual background being more likely to be motivated to meditate to achieve a spiritual experience, liberation, and enlightenment. Using the reasons to meditate provided by Sedlmeier and Theumer [1], we explored the link between reasons to meditate, PLEs, and meditation technique use.

## Mystical experiences

While the negative effects of PLEs have often been highlighted, it is noteworthy that people experiencing PLEs are also more likely to experience mystical experiences [31], and phenomenological overlaps between PLEs and mystical experiences have been identified [32, 33]. Both psychosis and mystical type experiences have been described as altered states of consciousness (ASC) [34]. An ASC is defined as a state where the neurocognitive mechanisms of consciousness are more likely to produce misrepresentations like hallucinations, delusions, and memory distortions. Examples of such states include dreaming, psychotic episodes, psychedelic drug experiences, certain epileptic seizures, and hypnosis in highly hypnotizable individuals [35]. The psychotic state of schizophrenia has been defined as an altered state of consciousness, whereby there are significant perceptual changes and alterations to meaning making [34]. Psychedelics have been used as a model of psychosis, but also found to produce mystical type experiences (measured via the Mystical Experiences Questionnaire) [33]. Experiences linked to the psychedelic state were found to correlate positively with aberrant salience, measured via the Aberrant Salience Inventory. These experiences included complex imagery (hallucination like), ego-dissolution and increased suggestibility [33]. Mystical experiences like this can be perceived as positive and have a strong relationship with meditation [36, 37]. For example feeling connected to "all of existence" or a profound positive mood among others [38] is described in relation to mystical experiences and can arise with the use of meditation [36, 37]. It has been suggested that mystical experiences, described as altered states of consciousness with changes to the experience of 'self', are considered beneficial in terms of insight and personal change [37, 38] for some. However, they can be more distressing for people with psychosis proneness, such as those diagnosed with schizophrenia [31] (see S25 'Supplementary Introduction' in S1 File for a detailed discussion). Here we measured mystical experiences using the Mystical Experiences Questionnaire (MEQ) and psychotic-like experiences with the Community Assessment of Psychic Experiences-42 (CAPE) positive subscale with three extra items from the Launay-Slade Hallucination Index (LSHI) [39].

## Cognitive processes in meditation

We also measured self-reports of cognitive processes previously suggested as core mechanisms of the beneficial impacts of mindfulness interventions [40]. Increased control over mind wandering has been theorised to link to better top-down regulation of somatosensory cortex and enhanced gain control, leading to better detection of when the mind drifts away from somatic attention [20]. The salience network which comprises of anterior insula cortex, anterior cingulate cortex and subcortical dopamine circuits has been found to show alterations in connectivity via MBCT practice [41]. This alteration was associated with more control, and sustained attention, in relation to body sensations [41]. This ability has been termed interoceptive awareness and can be thought of as the awareness of the 'global homeostatic state' of the body [42]. Interoceptive awareness includes awareness of changes in heart rate, gustatory sensations,

thirst and oxygen levels to name a few [43]. Improvements in facets of interoceptive awareness after mindfulness-based interventions have been linked to improvements in depressive symptoms, mediated by the ability to 'decenter' [44]. Decentering is described as the ability to observe thoughts and emotions without immediately being compelled to react and take a slightly distanced perspective on one's own experiences [45].

Interoception has also been found disrupted in Schizophrenia, where positive psychotic symptoms are a core feature [46]. Because of interoception's involvement with affective processing (feeling states), agency over actions and decision making, Yao and Thakkar [46] consider it a potential target for treating Schizophrenia, a disease that presents disruptions to the normal functioning within these domains. Some evidence already exists for utilising mindfulness-based interventions with Schizophrenia [47], that Yao and Thakkar [46] suggest might improve interoceptive awareness and symptoms.

## Study summary

To date, a clear link between different meditation methods and adverse experiences, in particular PLEs, is still missing. Here, we set out to test, in a sample of >600 meditators, whether meditation is associated with PLEs, and if so, which type of method is driving this relationship. We also measured self-reports of potentially protective cognitive processes interoceptive awareness [46] and decentering [44].

We hypothesised that meditation methods differed in their relationship with PLEs and participants' subjective ratings of how preventative or likely to increase PLEs they were (hypothesis 1). We focused on the positive symptoms of psychosis as our main PLE measure. We also hypothesised that meditation experience, such as months of lifetime meditation use and days on meditation retreat, would link to PLEs (hypothesis 2). We hypothesised that psychosis proneness would moderate the relationship between meditation techniques and PLEs (hypothesis 3). We also predicted that PLEs would positively correlate with mystical experience (hypothesis 4).

## Methods

### Design

This study used a repeated measures design employing an online survey. The first version of the survey took approximately one hour to complete with an option to stop halfway through. Using Qualtrics as the survey platform meant that even if participants stopped before the option to finish, their results would be saved once the survey timed out. The second version of the survey was shorter, to help increase the number of people completing it. We also had a follow up version to examine changes over time (only data from 30 were saved due to a technical error).

### Procedure

Participants were recruited either via social media channels associated with the authors, the University of Plymouth, or the Medito Foundation (link shared through Medito App). Author JS, ES and TP all shared a poster via their Twitter channels with details about the online study, as did the University of Plymouth psychology department. This poster was also shared via author ES' Facebook research lab channel. Participants were not paid for taking part in the study, but they could choose a charity to have money donated to on their behalf. 310 participants donated to the Medito Foundation, with the rest selecting a different charity. The recruitment started in February 2022 and stopped in September 2022 for the main survey, and

for the follow-up it began January 2023 and ended in March 2023. No analysis on this data occurred before a pre-registration was deposited on the Open Science Forum (OSF) July 2022, making this a pre-registered study [48]. Data from a pilot study was used to inform the construction of the scripts for analysis, but this pilot data was not included in the data set reported here.

## Participants

Our only inclusion criterion was a minimum meditation experience of one month. 613 meditators completed the survey up to the end of the meditation habits questions and the main PLE measure (CAPE-42), providing data to test hypothesis 1 (differences between meditation methods link to PLEs).

## Ethical approval and consent

The study was approved by the University of Plymouth, Faculty of Health, School of Psychology Ethics Committee (Ethics number: 2776), by email and letter through the online system. We started data collection after receiving ethical approval. Participants gave consent by ticking a box after having read the information sheet approved by the ethics committee.

## Materials and measures

The full survey had 20 question blocks, and 12 of these were validated questionnaires with 175 items in total. 6 blocks contained questions designed to capture data including demographics, meditation habits/experience (Lifetime use in months, regularity of practice a week, hours of meditation a week), meditation retreat experience, meditation method use, drug use and mental health, which totalled 65 questions. Two other blocks consisted of participant information with consent, and a block halfway through asking if the participant wished to continue. Participants were asked if they were currently meditating, and if they had stopped, they were asked to report when.

## Questionnaires

**Meditation questionnaires—Techniques, traditions, reasons to meditate.** To identify what meditation methods people had used, a list of 50 methods [19] (S1 in S1 File) were presented which used descriptions of what a meditator does when using that method. People were asked to select every method they had ever used before this list of endorsed methods was then presented again with a frequency of use rating scale: "Please rate all the techniques you selected on how often you use them from '0-Not at all', and '1-A Little' to '10-Very often'."

To gather data about which meditation traditions people identify with, or have identified within, a checklist was presented of traditions that were taken from the same study about meditation methods [19] (S2 in S1 File).

To understand the motivations for meditating amongst the participants, a list of reasons was included with a rating scale from 0–5. 13 of these were taken from a study by Sedlmeier and Theumer [1] on this topic, with a further two reasons being added by the researchers, which were: "To move towards a non-dual state of emptiness or oneness during a meditative state (Other terms: Buddha Consciousness, cosmic consciousness, pure consciousness, true-Self, non-Self)" and "Developing paranormal/psychic powers." Each reason was rated separately as a 'reason to start' and a 'reason to continue' (S3 in S1 File). For analyses, we added these scores.

**Clinical and psychological traits questionnaires.** The Community Assessment of Psychic Experiences-42 (CAPE-42) [15] is a 42-item questionnaire measuring psychosis symptoms, with subscales for positive symptoms (e.g., hallucinations), negative symptoms (e.g., anhedonia), and depression. In this study, questions were grouped into positive and negative/depressive symptoms. The scale uses a 1–4 rating for frequency, excluding the distress scale. Additional items from the Launay-Slade Hallucination Scale (LSHS) [39] were added for visual hallucination experiences. We analysed only the positive symptom scale with the additional LSHS items. The distress scale was removed, and 1 distress question was added at the end of each subscale from 0–4 to save time in filling in the questionnaire, and because the frequency score is deemed the most important in terms of assessing the prevalence of such experiences. We refer to the combined CAPE-42 and LSHS measure as CAPE-42+.

Mystical experiences have been associated with meditation [36], so the Mystical Experiences Scale [38] was used to capture these experiences, such as feeling connected to all living things, a sense of ineffability and alterations to the experience of time and space. The 30-item revised Mystical Experience Questionnaire (MEQ) used in this study consists of four subscales and was derived from a forty-three-item version [38]. The MEQ has four subscales including 'Mystical', 'Positive Mood', 'Transcendence of Time and Space' and 'Ineffability' (see supplements 'Mystical Experiences Scale' in S1 File for more details).

To measure interoceptive awareness, a trait found to be influenced by meditation [40], the Multidimensional Assessment of Interoceptive Awareness 2 (MAIA2) [49] was presented. The MAIA2 has 8 subscales which are 'Noticing', Non-Distracting', 'Not-Worrying, 'Attention Regulation', 'Emotional Awareness', 'Self-Regulation', 'Body-Listening' and 'Trusting' [49]. The Not-Distracting subscale has items such as 'I ignore physical tension or discomfort until they become more severe'. The Not-Worrying scale includes 'when I feel physical pain, I become upset' (Reverse scored). We removed the not-worrying and not-distracting scales because we wanted to reduce the overall length of our survey and decided out of the 8 subscales and 36 items that these were the least relevant to Interoception as we understand it. This is based on the fact these scales were aimed at measuring if people distract/worry about uncomfortable bodily sensations, and although relevant, perhaps arguably less relevant than the other six subscales. We acknowledge that without such constraints it is best to include all the subscales.

The Experiences Questionnaire (EQ) measures 'decentering'. The EQ assesses the ability to take a non-judgmental perspective on thoughts and feelings. It has a unifactorial structure with items scored from 0-Never to 4-All of the Time. Examples of items in the EQ are 'I am better able to accept myself as I am', 'I am consciously aware of a sense of my body as a whole', I can take time to respond to difficulties' and 'I view things from a wider perspective'. The highest score is 44, with higher scores indicating more decentering [45].

The Somatoform Dissociation Questionnaire-5 (SDQ-5) measures dissociative symptoms associated with the body, with five items scored from 1–5. Higher scores (up to 20) suggest a stronger propensity for this trait [50].

The Sleep Condition Indicator (SCI) is used to measure sleep quality. This eight-item scale rates various aspects of sleep on a five-point scale. Higher scores (up to 32) indicate better sleep quality [51].

The Hospital Anxiety and Depression Scale (Anxiety Subscale) was used to assess anxiety. Items include 'I get sudden feelings of panic', 'I can sit at ease and feel relaxed', and 'worrying thoughts go through my mind'. It is scored from '0-not at all' to '3-very definitely and quite badly' with a total possible score of 21, with higher scores indicating higher anxiety [52].

The Aberrant Salience Inventory (ASI) measures the significance placed on irrelevant stimuli. This 17-item scale has a scoring range of 0–5 for each item. It consists of one higher

order factor, with five lower order factors including 'Increased Significance' (F1), 'Senses sharpening' (F2), 'Impending Understanding' (F3), 'Heightened Emotionality' (F4), 'Heightened Cognition' (F5). Higher scores (up to 85) indicate more aberrant salience [53].

The Dissociative Absorption Scale (DAS) is an eight-item scale measuring the focus on internal stimuli, with scoring in 10% increments up to 80. Higher scores indicate a higher tendency to focus on internal experiences. Examples of items include 'Some people have the experience of sometimes remembering a past event so vividly that they feel as if they were reliving that event' and 'Some people find that when they are watching television or a movie they become so absorbed in the story that they are unaware of other events happening around them' [54].

The Sense of Agency Scale (SoAS) measures the extent of control over actions that people perceive they have. This 17-item scale has a scoring range from 1–7. It assesses 'Sense of Positive Agency' and 'Sense of Negative Agency', with higher scores (up to 91) indicating more perceived agency [55].

Six items from the Kundalini Awakening Scale (KAS) [56] were selected to capture experiences reported by meditators, such as those in Lindahl et al's study [26], that are not distinctly represented by psychosis measures such as the CAPE-42. This scale was scored from 1–7 (1-strongly disagree and 7-strongly agree) with a highest total of 42, with a higher score indicating more 'Kundalini Awakening' experienced by that individual.

**Additional measures.** To assess perceived causality, a scale was added to a selected number of psychometrics (CAPE-42+, Mystical Experiences, KAS, SoAS). We asked participants for a rating for each meditation method they had initially endorsed 'Have you found your meditation practice to either make these experiences less likely or more likely to happen?'. The likelihood scale ranged from "very unlikely" (-5) to 'very likely' (+5). Participants rated these perceived causalities for mystical experiences separately for both 'in daily life' and 'during/immediately after meditation'. For PLEs, they rated 'in daily life', and accidentally, the second column was phrased as 'Have you ever associated any of these experiences with the practice of meditation?'. Whether participants considered such experiences as positive or negative was assessed with an extra question. For distress we asked: "Thinking about the questions 1–23 you have just answered, please indicate how distressing you rated these experiences in daily life" with a scale from 0- ("Not Distressed") to 5- ("Very Distressed"). For positive ratings we asked: "Thinking about the questions 1–23 you have just answered, please indicate how positively you rated these experiences in daily life", with a scale 0- (Not Positive) to 5- (Very Positive).

## Analysis

**Meditation methods and categories.** We extracted from the question of 50 meditation techniques (see above) the amount of engagement with each of three categories (NDM, CDM, ADM) according to the Nash, Newberg and Awasthi's [22] taxonomy: In step one author TP categorised all of the fifty meditation techniques by placing them in one of the three categories. This was achieved by reading the description of what the meditator does when they use each method and judging whether this indicated an intention to move toward a specific meditative state. The examples given by Nash, Newberg and Awasthi [22] for methods in each category, and in Dahl et al's [23] classification system which has considerable theoretical overlap, were referenced with attention paid to specific characteristics of methods in each class. Some techniques are less obviously belonging to one category and therefore overlap with more than one. This initial list was presented to authors EF and JS, who commented and indicated agreement or disagreement for each category, before meeting with TP to discuss. We then produced a second list of meditations placed into the Nash categories that we all agreed upon. In the final

**Table 1. Demographics, meditation experience and PLE score.**

| Variable | Hypotheses 1.a, c and 2.a (n = 613) | Hypothesis 3 (n = 155) | Hypothesis 4 (n = 180) |
|---|---|---|---|
| Sex Male | 347 | 82 | 95 |
| Sex Female | 262 | 72 | 84 |
| Sex Prefer Not to Say | 4 | 1 | 1 |
| Mean Age (SE) | 35 (0.55) | 38.83 (1.2) | 38.35 |
| Mean Trauma Rating (SE) | 4.07 (0.11) | 4.22 (0.22) | 4.29 (0.22) |
| Immigration Status: Born in Country | 492 | 119 | 138 |
| Immigration Status: Lived Here <10yrs | 48 | 13 | 10 |
| Immigration Status: Lived Here >10yrs or Majority of Life | 73 | 23 | 13 |
| *Mean CAPE Pos 20 Items (SE) | 29.55 (0.3) | 27.48 (0.44) | 27.52 (0.44) |
| Mean Med Exp Months (SE) | 50.06 (3.1) | 55.24 (7.5) | 53.63 (7.28) |
| Participants Who Endorsed NDM | 353 | 85 | 101 |
| Participants Who Endorsed CDM | 594 | 154 | 178 |
| Participants Who Endorsed ADM | 551 | 143 | 167 |

*Note*: Summaries of key demographic variables, meditation experience and positive psychotic symptoms (*Clinical cut-off 50). Participants could complete the whole survey or only parts of the survey. For each hypothesis, the following number of datasets were available: H1a+H1c+H2a –n = 613 (Questions about meditation practice and PLE symptoms); H4: n = 180 (Included only those who completed the Mystical Experiences questions); H3a: n = 155 ("Psychosis proneness Qus", i.e. additionally completed questions about risk factors for psychosis). For the purposes of testing the pre-registered hypotheses, we considered a participant to have completed the whole survey to be those who completed it up to the end of the psychosis proneness measures, the last one being the Aberrant Salience Inventory (n = 155). For 'Participants Who Endorsed' this represents how many picked 1 or more technique relating to each meditation method. The mean score and standard error for the main psychosis symptom measure (CAPE-42 positive subscale) was 29.55 (0.3), well under 50, the clinical threshold to detect undiagnosed first episode psychosis [73]. 16 participants were above this threshold of 50. 'Immigration Status: Born in Country' is a measure of how recently people have moved to their country of residence.

step, five methods per category were selected by author TP, which were judged to be most distinctive and easily categorised, and to ensure each category had the same number of methods (Table 1). The same process was repeated as for step one, until agreement was reached, and the three categories finalised. We pre-registered this classification. The score for each category (NDM, CDM, ADM) was then computed as the summed endorsement of the five techniques making up each category. To categorize techniques according to Matko et al., we referred to Fig 2 in Matko et al. [25] which shows a 2D matrix of all techniques, along the axes of body orientation and activation. Techniques above each axis' zero point were labelled as 'high', otherwise as 'low' (see Table 2).

**Factor analysis for reasons to meditate.** We performed exploratory factor analysis with people's responses about how much they endorsed the fifteen reasons to start/continue meditating. We implemented factor analysis [57] using the R package "psych" [58]. The aim of factor analysis is to explain correlation between different items of a measure as a function of common underlying factors. The number of factors was determined using parallel analysis [59]. We used generalised least squares to fit the model. Factors were rotated obliquely (i.e. allowing correlations between the factors) to improve interpretability of the factors.

**Controlling for multiple comparisons.** To enhance statistical rigour, we pre-registered 4 hypotheses (total of 7 statistical tests—though given some problems with data collection, see section "Statistical analyses and deviations from pre-registration" in the S1 File, this was reduced to 5 tests that we could carry out). While we had not pre-registered multi comparison corrections across these 5 tests, we nevertheless now include a standard correction procedure [60] (Benjamini-Hochberg, S4 in S1 File).

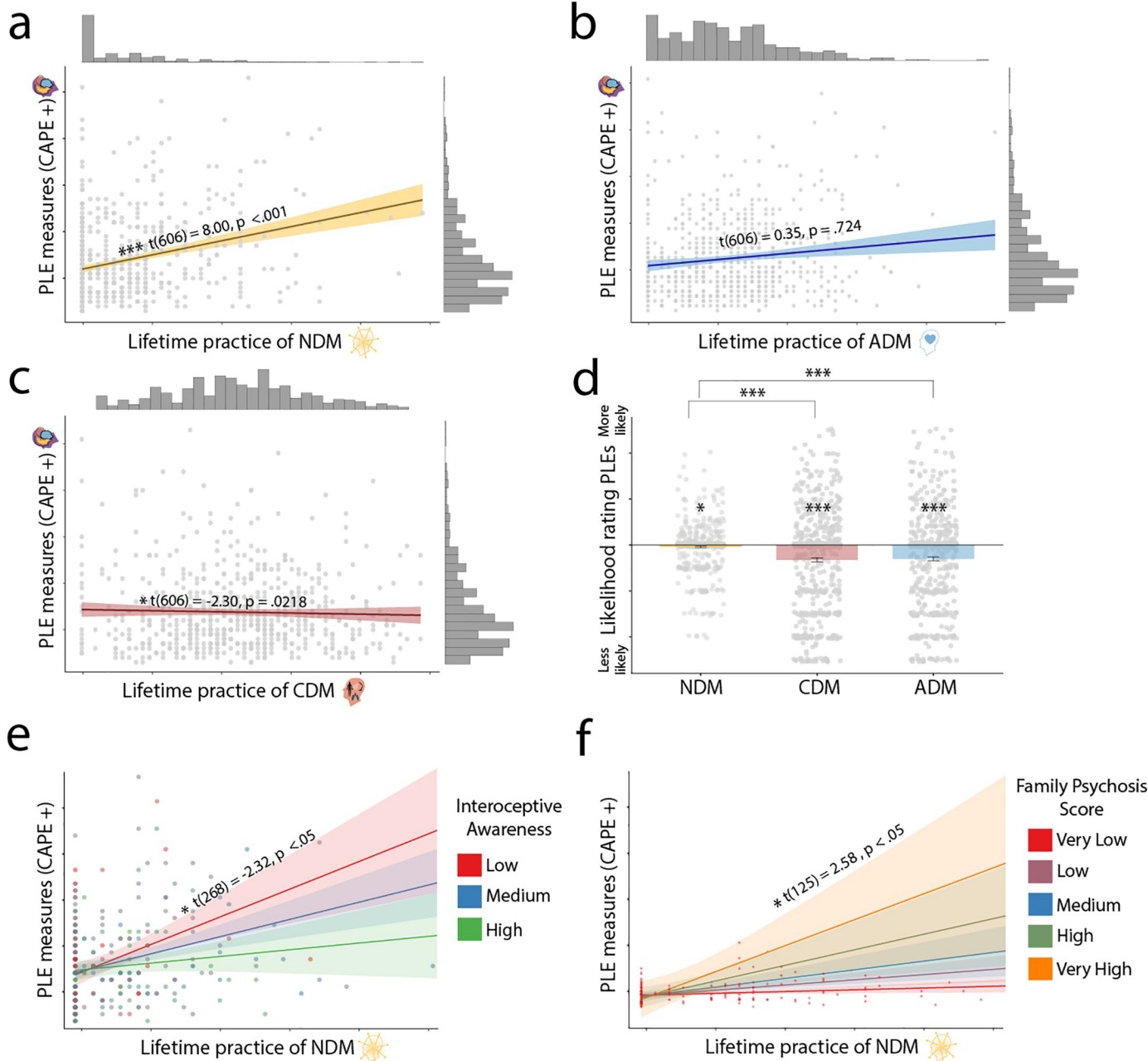

**Fig 2.** A-C. Relationship between lifetime use of NDM, ADM and CDM meditation methods and psychotic-like experiences (PLEs) in daily life from the regression output. While NDM correlated positive with PLEs, CDM showed a negative correlation. D. Participants average ratings for their perceived causality ('Likelihood rating') for each meditation category (difference from zero centre line, i.e. 'no relationship'). All techniques are rated as preventative ('less likely'), but with NDM being rates less so than CDM or ADM. E. Lifetime practice of NDM meditation and psychotic-like experiences in daily life with interoceptive awareness, measured by the Multi-Dimensional Assessment of Interoceptive Awareness 2, as a moderator: The lower interoceptive awareness, the stronger the link between NDM and PLEs. F. Lifetime practice of NDM meditation and psychotic-like experiences in daily life with family history of psychosis spectrum disorders, measured using a scale indicating if they have a relative diagnosed with a disorder (see methods for details), as a moderator: The higher the family history psychosis score, the stronger the link between NDM and PLEs. Significance stars relate to p value as follows: * = < .05, ** = < .01, *** < .001.

**Table 2. Techniques correlating with PLEs (during daily life), selected techniques for the three Nash et al. categories, and alternative classifications (Dahl an Matko).**

| | Coefficient | Nash Taxonomy | Dahl Taxonomy | Matko Taxonomy |
|---|---|---|---|---|
| **Top Positively Correlating Techniques** | | | | |
| (32) Opening oneself up to blessings and inspiration | 0.24 | NDM | D-ND | LB LA Affect Centered |
| (16) Contemplating a spiritually important question (e.g., "Who am I?") | 0.23 | NDM | D-ND | LB LA Contemplation |
| (33) Reading certain paragraphs of a text over and over again and taking them in | 0.22 | CDM | A-FA | LB HA Mantra/Contemplation |
| (41) Trying to feel one's heartbeat | 0.21 | CDM | A-FA | HB HA Body Centered |
| (38) Sitting and gazing at the wall, observing oneself doing nothing | 0.18 | NDM | D-ND | LB LA Mindful obs |
| **Top Negatively Correlating Techniques** | | | | |
| (7) Observing how bodily sensations arise without adhering to them | -0.17 | CDM | A-OMO | HB LA Mindful Obs |
| (2) Being mindful of the rise and fall of the abdomen while breathing | -0.11 | CDM | A-FA | HB HA Body Centered |
| (3) Observing how thoughts arise in the mind without adhering to them | -0.10 | CDM | A-OMO | HB LA Mindful Obs |
| (31) Observing emotions without adhering to them | -0.07 | CDM | A-OMO | HB LA Mindful Obs |
| (9) Being mindful of the sensations arising in the nose during inhalation and exhalation | -0.05 | CDM | A-FA | HB HA Body Centered |
| **Techniques used in Study** | | | | |
| **NDM** | | | | |
| (8) Singing sutras/mantras | 0.14 | NDM/CDM | A-FA/D | LB HA Mantra |
| (16) Contemplating a spiritually important question (e.g., "Who am I?") | 0.23 | NDM | D-ND | LB LA Contemplation |
| (20) Creating a visual representation of a deity and then merging with this visualization | 0.14 | NDM | C-P | LB LA Visual Concentration |
| (21) Droning or humming continuously with optional corresponding hand movements | 0.09 | NDM | A-FA | LB HA Mantra |
| (32) Opening oneself up to blessings and inspiration | 0.24 | NDM | D-ND | LB LA Affect Centered |
| **CDM** | | | | |
| (2) Being mindful of the rise and fall of the abdomen while breathing | -0.11 | CDM | A-FA | HB HA Body Centered |
| (4) Being mindful of the respiratory flow in the entire body | 0.03 | CDM | A-FA | HB HA Body Centered |
| (9) Being mindful of the sensations arising in the nose during inhalation and exhalation | -0.05 | CDM | A-FA | HB HA Body Centered |
| (24) Focusing on one point of the body and letting the breath flow through this point of concentration | 0.04 | CDM | A-FA | HB HA Body Centered |
| (25) Focusing on the pauses between inhalation and exhalation, carefully observing what happens | -0.00 | CDM | A-FA | HB LA Mindful Obs |
| **ADM** | | | | |
| (5) Perceiving, then releasing emotions and tensions (e.g., with the help of the breath), while scanning the body | 0.02 | ADM | A-FA | HB LA Affect Centered |
| (6) Cultivating compassion, sympathetic joy, equanimity, loving kindness (for oneself, friends, neutral people, enemies, the whole world) | 0.01 | ADM | C-R | LB LA Affect Centered |
| (26) Fostering and focusing on a spiritual connection created by singing together | 0.13 | ADM | C-R | LB HA Mantra |
| (37) Repeating an affirmation (e.g., "I am patient") | 0.14 | ADM | A-FA | LB HA Mantra |
| (50) With a specific intention (e.g., open one's heart, raise one's mood) selecting and repeating a mantra, combining it with associated hand postures or arm movements | 0.13 | ADM | C-R | LB HA Mantra |

*Note*: The 'coefficient' column gives correlation coefficients (Pearson's r) of each meditation technique with PLEs (see S26 'Supplementary Methods' section 'Meditation taxonomy correlations' in S1 File). For Dahl Taxonomy A = Attentional-Family, D = Deconstructive-Family, C = Constructive-Family, FA = Focused-Attention, ND = Non-Dual Oriented Insight, OMO = Open-Monitoring Object-Orientation, R = Relationship-Orientation, P = Perception-Orientation. For Nash Taxonomy NDM = Null Directed Methods, CDM = Cognitive Directed Methods, ADM = Affective Directed Methods. Matko Taxonomy HB = High Body Orientation, LB = Low Body Orientation, HA = High Activation, LA = Low Activation.

**Main analyses.** We structured our analyses hierarchically [61, 62] i.e. first performed a single statistical test for each pre-registered hypothesis. For most hypotheses, this meant a model comparison of two regression models, one containing the regressors of interest and one not containing them. This was then followed by statistical tests of the individual regression coefficients. This is conceptually analogous to an ANOVA followed up by post-hoc tests. Statistical software R [63] was used and packages required were dplyr [64], tidyverse [65], lme4 [66], effects [67], car [68], emmeans [69], sjplot [70] and ggplot2 [71].

When presenting the relationship of individual regressors with the outcome variable, we report regression coefficients, effect sizes (eta-squared, i.e. proportion of variance explained by the regressor of interest), t statistic and a p value. We checked assumptions of the regressions (see supplements 'Further Control Analyses—regressions' in S1 File). Code for analysis, and data, was made available (link here).

**Meditation method and PLEs (hypothesis 1a).** We used multiple linear regression to investigate the relationship between types of meditation (independent variables) and the main daily-life PLE measure (positive scale from CAPE-42 and 3 items about visual hallucination from LSHS [39]). The dependent variable was the sum score of the PLE measure (total score per participant on the CAPE-42+). The independent variables were meditation categories defined according to Nash et al [22] i.e. 3 categories. Pre-registered hypothesis 1b, i.e. the same test for PLEs during meditation could not be tested due to an error in phrasing the question (see supplements 'Statistical analysis and deviations from pre-registration' in S1 File).

$$PLEs \sim NDM\ Meditation\ Use\ +\ CDM\ Meditation\ Use\ +\ ADM\ Meditation\ Use\ +\ Age\ +\ Sex$$

**Perceived causality of meditation method use and PLE (hypothesis 1c).** We used linear regression models to predict the participants' likelihood ratings (-5 to 5), i.e. the perceived causality, of the use of different meditation techniques leading to PLEs. Techniques were categorised as described above.

$$Likelihood\ (of\ meditation\ leading\ to\ PLEs) \sim Meditation\ Category\ (NDM,\ CDM,\ ADM)$$

**Meditation habits, retreat experience and PLEs in daily life and during meditation (hypothesis 2).** We tested the relationship between PLEs and retreat experience, months of lifetime meditation experience and frequency of practice using linear regression.

$$PLEs \sim Retreat\ Days\ +\ Meditation\ Experience\ in\ Months\ +\ Meditation\ Hours\ Week$$
$$+\ Regularity\ of\ Meditation\ +\ Age\ +\ Sex$$

**Psychosis proneness, meditation and PLEs (hypothesis 3).** We tested whether psychosis proneness increased the strength of the relationship between meditation and PLEs using regression analysis. We predicted that psychosis proneness would moderate the relationship between meditation and PLEs.

$$PLEs \sim (NDM\ +\ CDM\ +\ ADM) * (Immigration\ Status\ +\ Sleep\ +\ DAS\ +\ ASI$$
$$+\ Trauma\ +\ Family\ Hist\ Psych)\ +\ Age\ +\ Sex$$

**Mystical experiences and PLEs (hypothesis 4).** We predicted that PLEs would positively relate to mystical experiences in 'daily life'.

$$PLEs \sim Mystical\ Exps\ +\ Age\ +\ Sex$$

Deviations from pre-registered hypotheses shown in the S26 'Supplementary Methods' in S1 File.

**Exploratory analysis.** Beyond our pre-registered analyses and related control analyses, we also explored other patterns in our data associated with 1) positive mental health traits (decentering and interoceptive awareness), 2) reasons for meditating, 3) Correlations of PLE and mystical measure items, and theoretical overlap.

**Exploratory analysis 1: Decentering, interoceptive awareness and PLEs.** We ran two regression analyses each with an interaction term. Firstly, we ran this for 'interoceptive awareness' using the MAIA sum score as the interaction term in a model with all 3 meditation methods in with PLEs as the outcome measure: PLEs ~ (NDM+CDM+ADM)*(MAIA Sum) + Age + Sex. The statistic of interest was the interaction effect of MAIA on each individual meditation method. Secondly, the same approach was used for the second regression, but instead we used the 'decentering' measure the EQ as the interaction term.

**Exploratory analysis 2: Reasons for meditating, techniques and PLEs.** We first tested whether reasons to meditate are differentially associated with the techniques people practised. We used a regression for each meditation category as the outcome variable and the two reason factors as the regressors. For example, for NDM: NDM ~ Reason Factor 1 (SpiritExplore) + Reason Factor 2 (Health) + Age + Sex. We reported the regression coefficients for each reason factor as the statistic of interest.

We next tested whether there was a relationship between why people meditate and reports of PLEs and mystical experiences daily, controlling for the techniques they practised. We used a model comparison to test a model with all three meditation categories (NDM, CDM, ADM) and the two reason factors 'SpiritExplore' and 'Health', with a model with only the meditation categories. Our statistic of interest was the ANOVA result for the model comparison.

$$PLEs \sim SpiritExplore\ +\ Health\ +\ NDM\ +\ CDM\ +\ ADM\ +\ Age\ +\ Sex$$

$$Mystical\ Exps \sim SpiritExplore\ +\ Health\ +\ NDM\ +\ CDM\ +\ ADM\ +\ Age\ +\ Sex$$

**Exploratory analysis 3: Correlations of PLE and mystical measure items (CAPE-42 (+3 LSH) and the MEQ), and theoretical overlap.** We used all the CAPE-42+ items and the MEQ items to create a correlation matrix (Pearson's r) to explore the links between individual items of our PLE mystical experiences measures.

**Post-hoc power analysis.** Post-hoc power analysis was conducted using the 'pwr' package in R [72]. We found that all our pre-registered hypotheses were adequately powered, meaning they all showed a power of > 0.8 (80%), the accepted general standard for power.

## Results

613 meditators completed at least parts of the survey (Table 1).

To investigate the relationship between lifetime use of different meditation methods and PLEs, we first categorised the meditation methods according to Nash, Newberg and Awasthi's taxonomy [22]. This resulted in the categorisation of three methods: Null Directed Methods (NDM), Cognitive Directed Methods (CDM) and Affective Directed Methods (ADM). Overall, we found that CDM is the most employed method, followed by ADM and NDM (Fig 1B,

Table 1). In terms of meditation traditions, the most popular selection was 'Use Meditation App' with 403 participants, while 385 said they had no tradition, 379 selected Mindfulness Based Stress Reduction, 238 'Yoga', 135 Mindfulness Based Cognitive Therapy and 100 'Theravada Vipassana'. Regarding the meditation experience of our full sample (n = 613) 54 had one to three months, 39 five to six months and the highest cluster at one to two years with 132 people (Fig 1E).

To compare different meditation taxonomies, we then also applied Dahl et al's (not pre-registered) and Matko and Sedlmeier's approaches to the 5 techniques selected for each category (see details of the Dahl et al., Nash et al. and Matko et al. frameworks in the introduction). Table 2 presents our comparison of different meditation taxonomies applied to techniques used in the study. The amount of body orientation the technique has can be observed in Table 2 by looking at the colour the technique is highlighted in (and abbreviation explained under table). The colour coding matches up techniques closely related to each other on 'body orientation', 'activation' and which cluster they seem most closely associated with in Fig 1 of Matko and Sedlmeier's article 'What is Meditation? Proposing an Empirically Derived Classification System'[24]. If a technique is low in body-oriented focus it will have one of two blue colours, but if high in body orientation it will have one of two red colours. For red 'high activation' techniques have the paler red with lower ones a darker red, and for blue techniques low activation have darker blue and higher lighter blue. In addition, we present the top 5 techniques that positively correlated with PLEs and those that negatively correlated with PLEs, applying the same approach to categorising them.

It is noteworthy that all techniques we classed as NDM [22] are classed as low in body focus [24, 74]. Contrastingly, all CDM techniques [22] were classed as high in body focus [74]. For ADM, these were mostly classed as low in body focus, with just one as high body focussed. Whilst all CDM techniques belong to the attentional family according to Dahl et al [23], no clear pattern emerged for the other two categories.

## Meditation method and PLEs in daily life

We tested whether meditation methods differed in their relationship to PLEs (hypothesis 1a). First, we related PLEs in daily life to participants' experience with each meditation category using linear regression (Fig 1C). Doing so, we found that a model with the use of different meditation methods was a significantly better fit to the data than a model with just age and sex and with no meditation use in (model comparison: $F(3) = 29.15$, $p = <0.001$) (Fig 2A–2C, S5 in S1 File). Specifically, our results show that NDM use is significantly and positively associated with PLEs ($t_{606} = 8.00$, $p = < .001$), CDM use was negatively associated with PLEs ($t_{606} = -2.30$ $p = < .05$) and there was no significant relationship between ADM and PLEs (S5 in S1 File). Effect sizes were calculated as partial eta-squared, and for NDM this was moderate, CDM small and ADM negligible (see S6 in S1 File for effect sizes).

To further explore Nash et al's, Dahl et al's and Matko et al's meditation categories and their links to PLEs, we first ran a Pearson correlation coefficient for all the meditation techniques and PLEs (see supplementary methods section 'Meditation taxonomy correlations' in S1 File). We then applied each of these categorising methods (taxonomies) to the top five strongest correlations in both the positive and negative direction (Table 2 bottom, 'Top positively correlated techniques' and 'top negatively correlated techniques'). We found that of the five techniques with the highest positive correlations to PLEs, 3 were rated as NDM [22], 3 as deconstructive, non-dual [23] and 4 as low body orientation [24]. For the five techniques with most negative correlations, 5 were rated as CDM [22], 5 as 'Attentional' [23] and 5 as having high body orientation [24].

## Perceived causality of meditation method use and PLEs

The analyses above establish correlations between the types of meditation and PLEs, but we also wanted to test perceived causality (hypothesis 1c, see Methods). There are different ways to assess causality (WHO 2013) [75], here we focus on a specific aspect, namely subjective attribution. To assess perceived causality, we asked participants to rate their perception of causality between the different meditation methods and PLEs (Fig 2D) using a scale running from -5 to 5 ('Have you found your meditation practice to either make these experiences less likely or more likely to happen?').

We found, indeed, that meditation techniques differed in their perceived causality. First, we related the meditation categories (NDM, CDM, ADM) to subjective ratings of how 'preventative' of PLEs the use of techniques within each category were. Doing so, we found that a model with the scores of how preventative each meditation category was differed significantly from one without the categories ($F(2) = 18.58$, $p < .001$, S4 in S1 File).

Post hoc analysis revealed that all three meditation methods were rated as preventative of PLEs in daily life (S7 in S1 File). However, our results also show that NDM was perceived as the least preventative method compared to the other two (Fig 2D, S7 and S8 in S1 File, NDM vs CDM: $t_{836} = -5.48$ $p < .001$; NDM vs ADM: $t_{836} = -5.04$ $p < .001$). As a control analysis, to test the perceived causality between meditation categories and PLEs only for those individuals who had experienced PLEs, we repeated the same analysis by only including meditators with at least 25% of PLEs scored as positive (n = 391; Mean CAPE positive score > 29). Our results showed that the perceived causality remained the same; the difference between NDM and CDM was significant ($t_{1818} = -5.33$, $p < .001$), as well as NDM and ADM ($t_{1818} = -4.86$, $p < .001$) (S9 in S1 File). All meditation methods also remained significantly different from zero (S10 in S1 File).

## Meditation habits, retreat experience and PLEs in daily life

We tested (hypothesis 2a; see Methods) whether overall lifetime exposure to meditation was associated with PLEs (lifetime meditation experience months, regularity of meditation, hours of practice a week, days spent on meditation retreat). To test this, we used a model with all of the meditation experience measures as regressors and PLEs as the dependent variable, versus an empty model with just age and sex. We did not find that lifetime exposure to meditation had a significant relationship to PLEs in daily life (model comparison: $F(2) = 1.11$, $p = 0.35$, S4 in S1 File). For completeness, we also report the results for individual regressors: days spent on retreat was not associated with PLEs ($t_{605} = 1.10$ $p = .27$), nor was total meditation experience in months ($t_{605} = -.01$ $p = .99$) meditation hours practise a week ($t_{605} = .75$ $p = .45$), or regularity of meditation practice a week/month ($t_{605} = -1.78$ $p = .08$, S11 in S1 File). The same result was found for separate regressions including each of the regressors individually (tested to ensure the absence of an effect was not due to collinearity between the lifetime exposure measures).

## Psychosis proneness, meditation and PLEs

We tested whether the relationship between types of meditation and PLE was moderated by psychosis proneness (hypothesis 3, see Methods). We captured psychosis proneness as a set of known risk factors for psychosis, namely environmental factors (sleep quality, immigration status and childhood trauma), cognitive characteristics (aberrant salience, dissociative absorption), and familial heritage (family history of psychosis and schizophrenia). To test for moderation, we compared a model containing interactions (i.e. moderating effects) between the three types of meditation and the psychosis proneness factors (S12 in S1 File), with a model

without these interactions (S13 in S1 File). We found a model containing the interaction terms was a better fit than one without (F(18) = 2.22, p < .01, S4 in S1 File).

Exploratorily, we looked post-hoc at the individual moderating effects. Family history of psychosis and schizophrenia moderated the relationship between NDM meditation and PLEs in a positive direction ($t_{125}$ = 2.58, p < .05, Fig 2F), i.e. in participants with higher family history score, there was a stronger relationship between NDM and PLE. Dissociative absorption, a trait previously associated with mental health conditions [54], showed a moderating effect upon CDM meditation in the negative direction ($t_{125}$ = -3.47, p < .001), and aberrant salience a moderating effect in the positive direction ($t_{1818}$ = 2.63, p < .01) which was also found for immigration status ($t_{125}$ = -2.19, p < .05) (how recently people have moved to country of residence) (S12 in S1 File).

We wanted to test whether people with a familial predisposition for psychosis were more likely to choose to use meditation methods classified as NDM. To achieve this our third regression analysis had three iterations, with one for each of the meditation methods as the dependent variable and familial history of psychosis spectrum disorders as a regressor along with sex and age. Familial history of psychosis spectrum disorders was not associated with any of the meditation methods with NDM ($t_{151}$ = -0.60, p = .55), CDM ($t_{151}$ = -.59, p = .56) and ADM ($t_{151}$ = .22, p = .82).

## Mystical experiences PLEs and meditation

Next, we investigated the associations between PLEs and mystical experiences. We tested whether mystical experiences were associated with PLEs in daily life (hypothesis 4a). We found that mystical experiences were positively associated with PLEs in daily life ($t_{175}$ = 3.76, p = < .001, S4 in S1 File).

Given that PLEs differentially associated with types of meditation techniques, we explored whether the same was true for mystical experiences. Indeed, NDM meditation use was associated with mystical experiences in daily life ($t_{149}$ = 4.85, p = < .001) (See Fig 3A–3C, S14 in S1 File) with CDM and ADM not reaching significance.

## Exploratory analyses

Beyond our pre-registered analyses and related control analyses, we also explored other patterns in our data associated with 1) positive mental health traits (decentering and interoceptive awareness), 2) reasons for meditating.

**Exploratory analysis 1: Decentering, interoceptive awareness and PLEs.** Decentering is a trait suggested to underlie the positive impacts upon mental health of trait mindfulness [36], and we wanted to test if it was preventative of PLEs, and if it moderated the relationship between NDM meditation use and PLEs.

We found that the more decentering ability people reported, the lower PLE symptoms ($t_{207}$ = -2.15, p < .05, S14 in S1 File). Decentering also interacted with NDM and ADM. Specifically, decentering moderated the relationship between NDM and PLEs in daily life ($t_{207}$ = -2.78, p < .01, Fig 3D): the lower the decentering ability, the stronger the link between NDM and PLEs. The interaction between decentering and ADM was also significant ($t_{207}$ = .3.13, p < .01), though note (see above) that there had not been a main effect for ADM in the previous analyses, making the interpretation more difficult. A similar finding was observed with interoceptive awareness, which has been described as the awareness of the global homeostatic state of the body [42], and measured in our study by the multidimensional assessment of interoceptive awareness 2 (MAIA2) [49]. We found a significant moderating effect in the negative direction for interoceptive awareness and NDM ($t_{248}$ = -2.32, p < .05, Fig 2E) and significant and

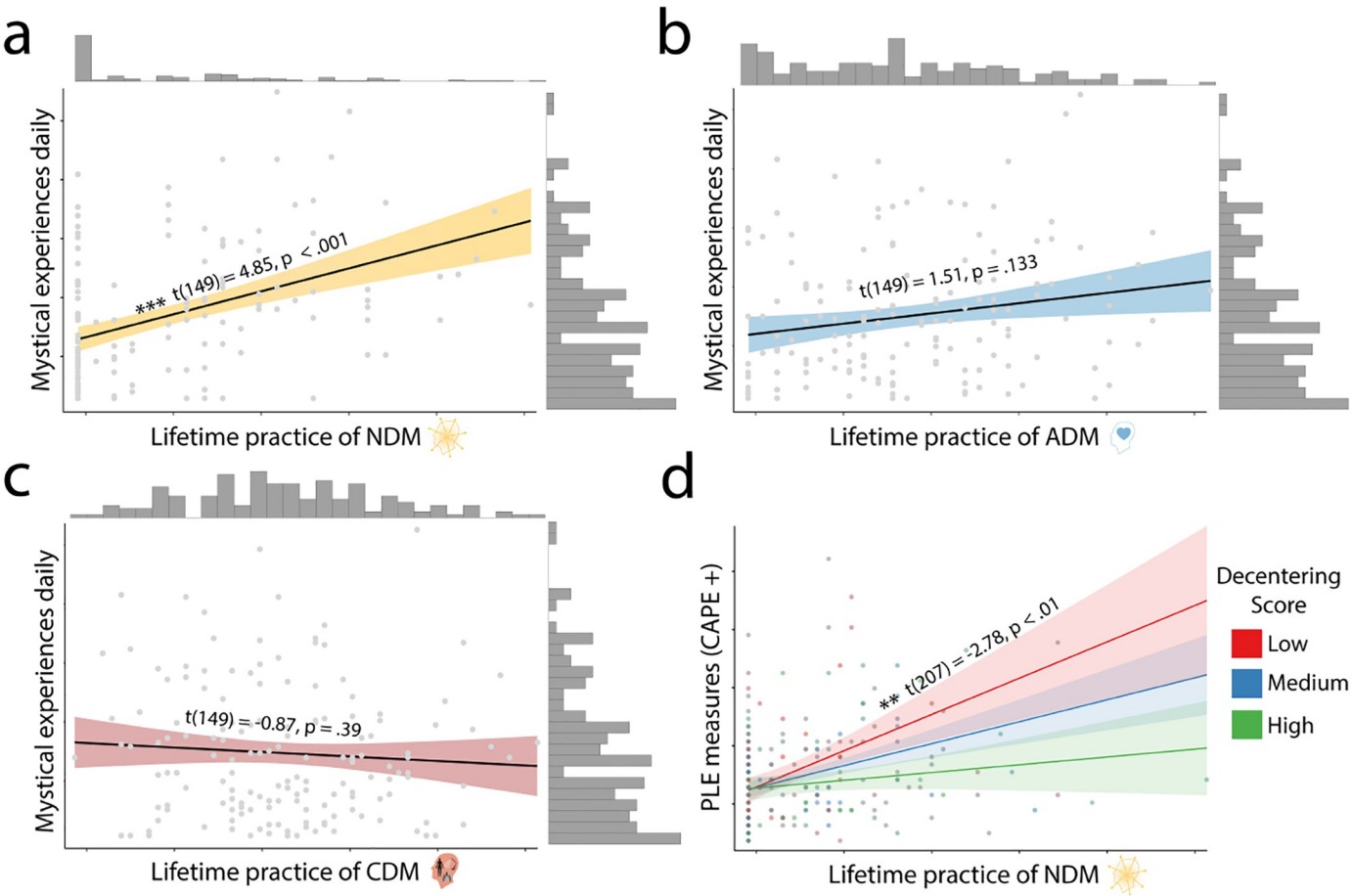

**Fig 3.** A-C. Relationship between lifetime use of NDM, CDM and ADM meditation methods and mystical experiences measured by the Mystical Experiences Scale [38] in daily life from the regression output. The higher lifetime practice of NDM, the more mystical experiences. D. Relationship between lifetime use of NDM methods and mystical experiences in daily life, with decentering measured by the Experiences Questionnaire [45] as a moderating factor. The lower the decentering, the stronger the relationship between NDM and PLEs. Significance stars relate to p value as follows: * = < .05, ** = < .01, *** < .001.

positive for ADM ($t_{248}$ = 2.78, p < .01) (S16 in S1 File). Further analysis revealed a significant positive relationship between CDM meditation use and interoceptive awareness ($t_{252}$ = 2.65, p < .01) and ADM meditation use ($t_{252}$ = 4.081, p < .001, S17 in S1 File) in daily life.

**Exploratory analysis 2: Reasons for meditating, techniques and PLEs.** We tested whether reasons for meditating associated with PLEs or mystical experiences. A factor analysis across our 15-item questionnaire identified two factors. Factor 1 comprises motivations associated with achieving a spiritual experience or exploring unconscious aspects of the mind (Factor 'SpiritExplore'). Factor 2 contains reasons pertaining to a motivation to achieve improvements in mental health, cognitive abilities, and connection to the body (Factor 'Health') (S3 in S1 File).

We first tested whether reasons to meditate are differentially associated with the techniques people practised. We found that the reason factor SpiritExplore was significantly positively associated with NDM meditation ($t_{607}$ = 8.93, p < .001, Fig 4A, S18 in S1 File). SpiritExplore was significantly negatively associated with CDM meditation ($t_{607}$ = -2.36, p < .01, S19 in S1 File) whilst reason factor Health was positively associated with CDM technique use ($t_{607}$ = 4.96, p < .001, Fig 4B, S19 in S1 File). Both factors were positively associated with ADM technique use (Health factor: $t_{607}$ = 5.70, p < .001; SpiritExplore: $t_{607}$ = 2.57, p < .01, S20 in S1 File).

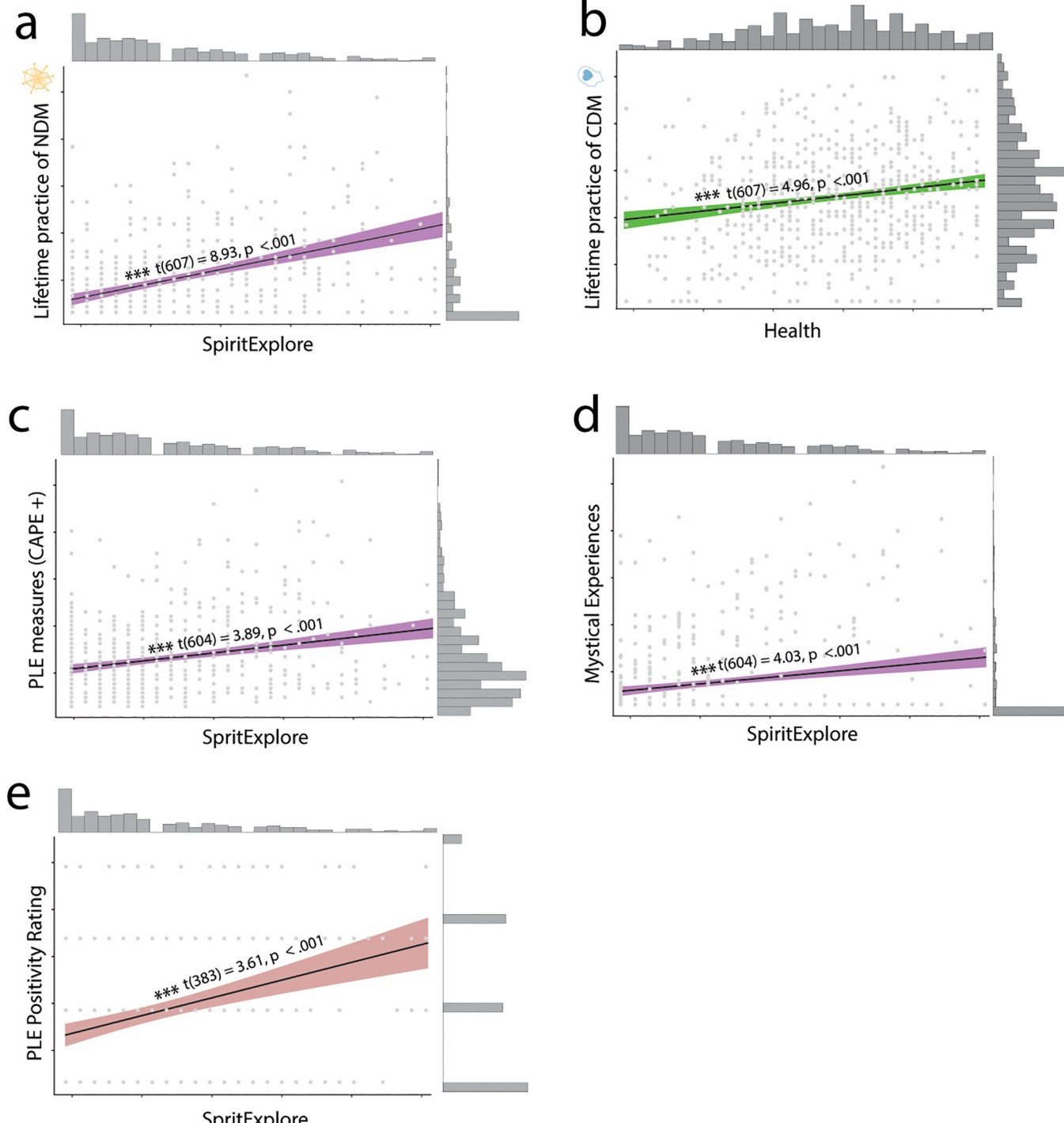

**Fig 4. Relationship between reasons to meditate, meditation practices and PLE and mystical experiences.** The more people reported starting to meditate for spiritual reasons, the higher the lifetime practice of NDM (A), the higher also the PLEs (C) and mystical (D) experiences during daily life, as well as the positivity ratings for PLEs (E). The more participants reported starting to meditate for health reasons, the higher the lifetime practice of CDM (B). Significance stars relate to p value as follows: * = < .05, ** = < .01, *** < .001.

We next tested whether there was a relationship between why people meditate and reports of PLEs and mystical experiences in daily life, controlling for the techniques they practised. We used a model comparison to test a model with all three meditation categories (NDM, CDM, ADM) and the two reason factors SpiritExplore and Health, with a model with only the meditation categories. We found that for PLEs, the model with the two reason factors was a better fit to the data than the one without (model comparison: $F(2) = 11.48$, $p <0.001$, S21 in S1 File). Specifically, SpiritExplore was positively associated with PLEs ($t_{604} = 3.89$, $p < .001$, Fig 4C, S22 in S1 File) while Health was not. For mystical experiences, the model with the reason factors was likewise a better fit (model comparison: $F(2) = 8.17$, $p = <0.001$, S21 in S1 File). SpiritExplore was positively associated with mystical experiences ($t_{604} = 4.03$, $p < .001$, Fig 4D) but Health was not ($t_{604} = -1.50$, $p = .13$). The meditation categories remained significant, with NDM positively associated with PLEs and CDM in the opposite direction (S26 in S1 File), suggesting that both type of meditation and reason for meditating have partially independent relationships to PLEs and mystical experiences.

**Exploratory analysis 3: Correlations of questionnaire items from the CAPE-42+ and the MEQ.** To try to understand better the link between PLEs and mystical experiences, we ran an exploratory analysis at the item level between the two questionnaires (CAPE-42+ and MEQ, see S27 'S1 Fig' in S1 File). First, we note that in the CAPE-42+, out of twenty-three items, 5 contain negative connotations (see below), while in the mystical experiences questionnaire, six contain positive connotations, all from the 'positive mood' subscale. The other items do not have clear positive or negative connotations.

1. Do you ever feel as if you are being persecuted in some way?

2. Do you ever feel as if there is a conspiracy against you?

3. Do you ever feel that people look at you oddly because of your appearance?

4. Have your thoughts ever been so vivid that you were worried other people would hear them?

5. Sometimes a passing thought will seem so real that it frightens me.

We first examined the correlations between the CAPE-42+ and the mystical subscales across participants. We found that the large majority of MEQ subscale sum scores were significantly and positively correlated with CAPE-42+ items (except five items, see S1 Fig in S1 File). More specifically, the majority of CAPE-42+ and MEQ items showed positive correlations, with the top five ranging from 0.29 to 0.37 (S23 in S1 File). The highest positive correlation was between the CAPE-42+ item "Do you ever feel as if a double has taken the place of a family member, friend or acquaintance?" ('Capgras delusion' [15]), and MEQ item from the space/time subscale ("Being in a realm with no space boundaries", Pearson's $r = 0.37$. MEQ item "Sense of being "outside of time, beyond past and future" from the same subscale also associated with the Capgras item ($r = .33$). MEQ item from the 'mystical' subscale ("Experience of oneness or unity with objects and/or persons perceived in your surroundings") was likewise associated with this Capras item $r = 0.26$. It could be suggested there is some overlap between these items in terms of alterations to the perceptions of boundaries, for example between self and other. It is worth noting that positive correlations exist between many items that seem to share little phenomenology (e.g., CAPE item: "Do you ever feel as if things in magazines or on TV were written especially for you?" and MEQ item: "Experience of oneness in relation to an "inner world" within.", $r = 0.27$). The most notable anti-correlation was found with CAPE item 1 about persecutory ideations ("Do you ever feel as if people seem to drop hints about you or say things with a double meaning?"). This item was negatively correlated with the top

five items from the 'positive mood' subscale of the MEQ with correlations ranging from -0.14 to -0.24. In summary, we conclude there are items that can be argued to share some phenomenological overlaps, but most do not have obvious themes that overlap. This is addressed further in the discussion section.

## Discussion

In a pre-registered large online survey (n = 613), we tested whether distinct types of meditation methods differentially correlated with psychotic-like experiences (PLEs) and mystical experiences. We first applied a previous taxonomy based on the goal states of meditation methods [22] and grouped meditation methods [24] into Null Directed Methods (NDM), Cognitive Directed Methods (CDM) and Affective Directed Methods (ADM). We found that lifetime use of NDM methods, which aim at achieving an altered state of consciousness devoid of phenomenological content [22], correlated with PLEs as measured by the CAPE-42 positive symptom scale (plus three additional items). Lifetime use of CDM methods showed a negative and statistically significant relationship with PLEs in daily life. Although cross-sectional survey data can't prove a causal relationship between meditation type and PLEs, we have found that meditators perceive that different types of meditation may be more likely to elicit PLEs than others. Indeed, they rated CDM and ADM methods preventative of PLEs, and while NDM was also rated as preventative overall, it was rated as significantly less so than CDM or ADM. This held true for the whole sample (n = 613), as well as for a sample including only those scoring 25% or above on the mean PLE score. Taken together, these findings suggest that the use of meditation methods with a focus on attaining an 'empty state' devoid of phenomenological content correlates with an increase in PLEs, while methods aimed at an enhanced cognitive state showed the opposite correlation.

### NDM techniques and low body focus correlate with PLEs

Although there is no general agreement about how to categorise meditation techniques, when applying three approaches to categorise the techniques correlating either positively or negatively with PLEs, we found that amount of body focus was a common factor that cut across them. The positively correlating techniques lean towards less body focus, whereas the negatively correlating techniques lean towards higher amounts of body orientation. We interpret Nash et al's [22] 'null' element of NDM methods to indicate an aim to achieve an empty state, which includes the absence of a focus upon interoceptive signals representing the 'homeostatic state of the body' and self-awareness [42]. CDM on the other hand, includes techniques that aim to enhance awareness and focus upon an 'object', which might include the body. A similar 'body scan' technique to the one that we used in this study has previously been associated with improved interoceptive awareness [76]. It might be that amount of body focus in relation to a meditation technique could influence factors related to PLEs. Disruptions to the experience of the bodily self [77, 78], and interoception have been associated with PLEs [46]. Although there is little in the way of direct studies on the link between NDM techniques and interoception, prior work investigating neural networks indicates brain regions linked to interoception demonstrate disruptions during NDM meditation [79].

### The link between NDM and PLEs is moderated by psychosis proneness and cognitive mechanisms

Psychosis proneness moderated the relationship between NDM meditation and PLEs, with people with higher psychosis proneness showing a stronger link between NDM and PLEs. Previous studies have reported that people with a higher propensity to psychosis are more likely to

experience hallucinations during sensory deprivation [80]. Psychosis proneness has also been associated with a proneness to psychotic episodes triggered by meditation by a number of case reports [9], and it has been suggested that meditation might lead to hallucinations such as seeing lights, through the same mechanism by which sensory deprivation does [81]. Lindahl et al [81] proposed that both sensory deprivation and meditation attenuate sensory input, leading to spontaneous firing of neurons via homeostatic plasticity.

## NDM techniques also correlate with mystical experiences

Our results showed that NDM meditation was associated with both PLEs and mystical experiences. We also found that PLEs correlate strongly with mystical experiences, supporting previous findings [31]. In exploratory analyses, we found this to be true even on the level of individual items. The strongest links were found between the CAPE-42+ item capturing perception of relatives being replaced by an imposter ("Do you ever feel as if a double has taken the place of a family member, friend or acquaintance?") and several mystical items associated with a change in the perception of spatial boundaries including those associated with the self (e.g. "Being in a realm with no space boundaries" or "Experience of oneness or unity with objects and/or persons perceived in your surroundings"). We also found correlations between items with less obviously shared phenomenology. Together, these results suggest that a shared underlying trait is captured that can express itself in symptoms more commonly described as psychosis-like and others more commonly described as mystical experiences. However, we cannot exclude here that positive correlations are associated with response biases in the questionnaires and further work is needed (e.g., designing questionnaire with reversed items, performing in depth phenomenological interviews).

## Meditating for spiritual reasons correlates with PLEs

We explored links between reasons people meditate and PLEs as well as meditation techniques used. We found that reasons associated with seeking spiritual experiences ('SpiritExplore') were positively associated with NDM meditation and with PLEs, but negatively associated with CDM meditation. Conversely, reasons associated with health ('Health') were positively associated with CDM but not with NDM and negatively with PLEs. For ADM both factors (Health and SpiritExplore) were significantly positively related. This suggests that the reasons people meditate are not only associated with the techniques they engaged with, but also to the presence of psychotic-like experiences. Further investigation is needed to better understand if there are underlying traits linked to reasons to meditate and PLEs. For example, further investigation could seek to clarify the mechanism linking a desire to have a spiritual experience and PLEs. Prior evidence has linked proneness to psychosis with an acceptance of beliefs in telepathy, miracles and mystical events measured via the 'Spiritual Acceptance Scale' [82], and toward a tendency for religious themes related to psychosis [83].

## Decentering and interoceptive awareness

Although we found a link between psychosis proneness, meditation and PLEs, people with no personal or family history of psychosis have also reported experiencing a psychotic episode in relation to meditation [9, 10]. This finding raises the question as to whether other traits in the population might moderate the relationship between the use of certain meditation methods and PLEs, such as those that have been shown to protect against mental ill health. Decentering has been suggested to be the main mediating factor between trait mindfulness and its benefits to mental health [84]. Our findings support this idea and suggest that the ability to 'decenter' provides a protective factor against PLEs. The aim to develop this ability has been associated

with meditation, in particular those seeking to develop a state of non-judgemental awareness of thoughts and feelings [23]. It should be noted that this trait is considered a marker of good mental health and associated with positive therapeutic outcomes [40], therefore it is not necessarily clear which direction the relationship might work regarding meditation based on survey data alone.

Our findings support the idea that decentering and interoceptive awareness are associated with mental health by showing that both moderate the effect of NDM meditation use upon PLEs, with the caveat that our measure has limitations due to it being a self-report measure that may be considered to measure 'interoceptive sensibility' rather than 'interoceptive accuracy' or 'interoceptive awareness' (Discussed in limitations section). In relation to this discussion, it is relevant to note there is evidence that interoceptive awareness is disrupted in schizophrenia and can be characterised as aberrant predictive inference [46]. Such a finding is suggested to provide potential methods for treating psychotic symptoms [46], such as using mindfulness based interventions which have been shown to have some success in reducing positive psychotic symptoms in schizophrenia [85].

In summary, our findings suggest that a more generally positive experience, and increased awareness, of the body moderates the relationship between meditation techniques and PLEs.

## Limitations

Meditators selected the meditation techniques they practise from the list of fifty descriptions of techniques, validated in a previous study [24], without needing to provide further explanations about their goals using each technique or interpretations of these techniques. Individuals may have differed on how they interpreted each technique. For example, one person may select a technique believing it to relate to a specific feature of their tradition, but another may interpret the same technique as one they use in a different tradition. Although this is something to consider, it should be noted that it is possible that a description of what someone does using a specific technique within a specific tradition is the same as a technique with a different name from another tradition.

We assessed correlations between types of meditations and PLEs, as well as perceived causality based on participants' subjective reports of causality. Risk factors (which temporally precede the onset of meditation) were captured as family history of psychosis. However, In the absence of longitudinal data, we cannot draw conclusions about the temporal sequence of interactions between meditation techniques and PLEs. Future work should use a longitudinal design or an intervention study.

It is also worth considering, as with any regression analysis using survey data, that unmeasured factors may be contributing to the perceived links between regressors and the independent variable despite our efforts to reduce this possibility. Regarding the sample used to test our hypotheses, they were all self-selecting meditators. There may be factors relating to 'self-selection' and to the fact over half were recruited via a meditation app that reduce the generalizability to the general population, and to the meditation population. The other participants were recruited via Facebook and Twitter, which might bias towards a population that are savvy with social media. We purposefully allowed for a wide range of experience with meditation. We also acknowledge that although we did not collect data about ethnicity, nationality, sexual orientation or first language, that these factors might be considered relevant by some people and could be considered for future studies.

The questionnaires used in our study were drawn from previous validation studies. Our PLE measure, the CAPE-42+, is a widely used measure of experiences commonly associated with the risk of developing psychosis. It might be considered that there is a bias towards

'unpleasant' or 'unwanted' experiences with this measure, for example with items measuring 'paranoid ideation' (and 4 others). Vice versa with the mystical experiences measure (MEQ), there is a bias towards positive experiences, for example with a subscale dedicated to positive experiences. The measure for decentering (EQ) measures a trait considered beneficial for mental health. Thus, although our measures were carefully selected, it should be taken into account the inherent bias towards positive or negative experiences that each may lean towards. To understand what different scores on each scale mean, it is advised to read the methods section for a more detailed account. The measure we used to test interoceptive awareness (MAIA2) leans towards gathering data on positive experiences of the body. Therefore it might be considered an indirect measure of this trait, and similarly it is a self-report measure rather than a physiological test like the 'Cardioception'[86] experiment of heartbeat discrimination, so should be considered a test of interoceptive sensibility rather than accuracy [87].

## Supporting information

**S1 File. Updated supplements with all supplementary tables, introduction, methods, and figure.**
(DOCX)

## Author Contributions

**Conceptualization:** Timothy Palmer, Jacqueline Scholl, Elsa Fouragnan.

**Data curation:** Timothy Palmer.

**Formal analysis:** Timothy Palmer, Jacqueline Scholl, Elsa Fouragnan.

**Funding acquisition:** Elsa Fouragnan.

**Investigation:** Timothy Palmer.

**Methodology:** Timothy Palmer, Kenza Kadri, Eric Fakra, Jacqueline Scholl, Elsa Fouragnan.

**Supervision:** Jacqueline Scholl, Elsa Fouragnan.

**Visualization:** Timothy Palmer.

**Writing – original draft:** Timothy Palmer.

**Writing – review & editing:** Timothy Palmer, Kenza Kadri, Eric Fakra, Jacqueline Scholl, Elsa Fouragnan.

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
