## [Decision Letter · Decision Letter 0]

20 Jul 2023

PONE-D-23-11057Differential Relationship Between Meditation Methods and Psychotic-Like and Mystical ExperiencesPLOS ONE

Dear Dr. Palmer,

Thank you for submitting your manuscript to PLOS ONE. After careful consideration, we feel that it has merit but does not fully meet PLOS ONE’s publication criteria as it currently stands. Therefore, we invite you to submit a revised version of the manuscript that addresses the points raised during the review process.

Please address all the comments and questions raised by the reviewers. As per the request of one of the referees, please have a statistician review the analyses to ensure the robustness of the statistical methods 

We look forward to receiving your revised manuscript.

Kind regards,

Rakesh Karmacharya, MD, PhD

Academic Editor

PLOS ONE

Journal Requirements:

3. We note that Figure 1 in your submission contain copyrighted images. All PLOS content is published under the Creative Commons Attribution License (CC BY 4.0), which means that the manuscript, images, and Supporting Information files will be freely available online, and any third party is permitted to access, download, copy, distribute, and use these materials in any way, even commercially, with proper attribution. For more information, see our copyright guidelines: http://journals.plos.org/plosone/s/licenses-and-copyright.

Reviewers' comments:

Reviewer's Responses to Questions

**Comments to the Author**

1. Is the manuscript technically sound, and do the data support the conclusions?

Reviewer #1: Partly

Reviewer #2: Yes

2. Has the statistical analysis been performed appropriately and rigorously? 

Reviewer #1: I Don't Know

Reviewer #2: Yes

3. Have the authors made all data underlying the findings in their manuscript fully available?

Reviewer #1: Yes

Reviewer #2: Yes

4. Is the manuscript presented in an intelligible fashion and written in standard English?

Reviewer #1: Yes

Reviewer #2: Yes

5. Review Comments to the Author

Reviewer #1: This is a very interesting but under-researched area of study, and these results have potential clinical implications, so it is welcome that the researchers have decided to conduct this study. Below I describe my main concerns and suggestions.

Causal inference

In the section “Subjective Causality of Meditation Method Use and PLEs” you talk about a criterion for causality citing WHO 2013, “an event that happens concurrently or right after an intervention.” This addresses one of many causal attribution criteria for assessing adverse effects of pharmacological interventions, but it is not really sufficient on its own – please reflect this in your wording. The Varieties of Contemplative Experience paper discusses causal attribution criteria in greater detail (see https://doi.org/10.1371/journal.pone.0176239 page 28), and subjective judgement of causality is just one of eleven factors used in that study to determine causality.

In discussing causality for this study it is overstretching to talk about a causal relation between meditation method and PLEs. It would be good to consistently use language like “perceived causality” should be used to discuss the finding that participants felt that a method of meditation was causing the experience of a PLE. Additionally, these questions should not be asked of participants who have not had a PLE, because their inclusion in the data skews your results further toward hypothetical perception of a relationship, rather than an empirical one, derived from first-hand experience of those who have had one. You also imply that this relationship is a fact of which participants are subjectively aware in the first paragraph of the discussion. There is not sufficient evidence for this claim.

Operationalise Your Terms

It would be good to clearly define, for the sake of the study what definitions you’re using for mystical experiences and PLEs. You do explore the idea that PLEs are the positive aspects of psychosis in a parenthetical statement in your intro, but less discussion is given to what a mystical experience is—perhaps the working definition of both terms is the same and you’re curious how survey participants define them differently. Add a paragraph or two explicitly operationalising your key terms, what working definition you used, and how you measured it (in addition to your later section that describes each of the scales). This is done somewhat in the section called “Differences Between Mystical Experiences in Subjects Distress and Positive Ratings,” but not sufficiently. Are there positive and negative mystical experiences, or have you defined a mystical experience as a positive PLE?

There is not sufficient treatment of the PLEs at the same time being desirable and a source of insight on the one hand, and on the other, constituting a risk against which certain methods of meditation constitute a protective factor. It would be appropriate to address why some people are seeking PLEs/mystical experiences, and some people may wish to avoid them.

Document Layout

Please list your hypotheses more clearly at the end of the introduction (as you have done in the pre-registration).

The number of participants that completed the survey is part of the results, in the methods section please explain the study participant selection criteria (eg how did you define “meditator” – was there a minimum required amount of practice?).

What did you use the motivation to meditate data for? Please clarify.

Have your study design precede the description of participants and scales rather than follow them. Also, please clarify what happened to the follow-up questionnaire. You don’t seem to report any results on it.

Limitations

There is no discussion of the limitations of this study. In your discussion section you talk about the implications and potential application of findings, but there should also be a paragraph that talks about the limitations of causal inference especially in relation to the type of study (cross-sectional survey) and its claims.

You have conducted several statistical analyses, and have not controlled for multiple testing (false discovery rate). Therefore, some of the associations that you found may be spurious, which needs to be acknowledged as a limitation. This is the most important limitation of your work, so please mention it both in the text and in the abstract.

You should also address the degree to which the categories of meditation method are arbitrary and may overlap or be mutually exclusive, as well as any efforts made toward inter-rater reliability in classification of meditator’s self-description of their meditative goals and techniques, as it appears this may have only been done by author TP. Were there any limitations due to your sample population or approach to recruitment? You state that 613 participants completed at least parts of the survey. How many completed the whole survey, and are there any limitations due to the number that completed it only in part?

If you have collected your data before registering your analysis plan, unfortunately this does not technically count as pre-registered (even if you have not analysed your data before registration). Please acknowledge this as a limitation.

You wanted to recruit 1000 participants in order to have a large enough sample size to have participants address 50 different techniques, are there any limitations in the fact that you recruited just over 600? Did you have to make any adjustments in the study design to accommodate this change in sample size?

Also, the Dahl classification was not preregistered, please acknowledge this.

Statistical Analysis

It is not clear why you have used the model comparison technique rather than simply looking at the beta coefficients of the predictors of interest in the full multiple regression model. And what statistical test do the t values reported correspond to (ie exactly what are you comparing)? I am not sure you are actually testing what you want to test. Looking at the beta coefficients of your predictors would be a much more direct way (and possibly the only correct way – I am not a statistician) to test your hypotheses.

You mention that you are not presenting all your results (eg other questionnaires) due to constraints on word count, but PLOS ONE does not have any word count constraints, so please include all your results.

Reviewer #2: I appreciate the opportunity to review the manuscript under consideration for publication entitled “Differential relationships between meditation methods and psychotic-like and mystical experiences.” The manuscript is well written and provides novel and important findings for the relationship between meditation practices and psychotic-like experiences. The grouping of meditation practices into different taxonomies is an especially strong characteristic of the study, and allows for a more nuanced picture of the relationship between types of meditation practices and altered state experiences. I am not aware of any prior publications that have conducted such an analysis. Furthermore, the analytic plan is strongly designed and the results are convincing. I believe that this paper provides important information that could be used to identify and monitor risks for adverse effects of meditation. As a result, I recommend this paper for publication considering that the authors address the concerns outlined below:

1. My primary concern is that the manuscript presents an overly simplistic narrative about the relationship between mystical experience and psychotic-like experiences. For example, the discussion in the second to last paragraph of the introduction on the relationship between mystical experience and psychotic-like experiences could benefit from a more critical analysis of the construct of mystical experience (see Taves, 2020). Mystical experience and psychotic-like experiences both represent altered states; however, the mystical experience scale conflates these states with positive affect and positive appraisals, while the measure of psychotic-like experiences that this manuscript uses likely does the opposite. Therefore, it is unsurprising that these two measures of altered states are related to each other (as the results found). Furthermore, differences may be more representative of whether these altered states co-occur with positive or negative affect and/or interpretations (though phenomenology may vary between the scales too). I would like to see more discussion of the similarities and differences (in terms of phenomenology, affect, interpretation) between mystical experience and psychotic-like experience regarding both the general constructs and the specific measures used.

Taves, A. (2020). Mystical and Other Alterations in Sense of Self: An Expanded Framework for Studying Nonordinary Experiences. Perspectives on Psychological Science 15(3), 669-690. https://doi.org/10.1177/1745691619895047

2. I would also like to see more description of the interoception and decentering scales used in the manuscript. Given that these scales were designed to measure adaptive psychological skills, I think it is important to note that they were not designed to be neutral in terms of mental health but instead to measure inherently healthy processes. For example, the MAIA contains the item “I feel that my body is a safe place.” While I do not think that this presents a problem for the present findings, I do think that it is important to note that that these scales are not measuring body awareness and cognitive distancing as unbiased meditation-related processes, but as inherently psychologically healthy skillsets in relation to such processes.

3. For all measures, it would be helpful to include a more thorough description (e.g., subscales, total score) and example items to illustrate what the measure is capturing. I believe that this is important given that the results are dependent on the measures used.

4. When reporting regression results, I would like to see some measure of effect size, such as standardized beta.

5. The columns in Table 1 (during meditation QUs and Psychosis Proneness QUs) are hard to follow without reading the methods section in detail, which is provided much later in the manuscript. I am still not sure what QU stands for and would like to see this defined in the table note. I would also like to see the three samples described in more detail in the table note, as well as how these differing number of participants relate to the analyses conducted.

6. I would also like to see more description of table 2, either in the table note or the results section text, as I find this table hard to follow as well. Please specify exactly what the colors mean and how the rows are organized. It would be helpful to specify that “top positively correlating techniques” refers to correlation with PLEs. I am still somewhat confused by what “techniques used in study” refers to exactly (weren’t all of the techniques used in the study)?

7. Finally, the description of the interaction results is confusing. The sentence “Our results showed interaction between NDM and PLEs in daily life was significantly negative (t207 = -2.78 p = <.01), and positive for ADM and PLEs (t207 = .3.13 p = <.01) (S1 Table 9, Fig. 3D)” seems to be reporting a moderation analysis; however, I think more information is needed to understand this statement. I believe that moderation would involve three variables but I only see mention of two variables in this sentence.

6. PLOS authors have the option to publish the peer review history of their article (what does this mean?). If published, this will include your full peer review and any attached files.

Reviewer #1: No

Reviewer #2: **Yes: **Nicholas K Canby

---

## [Author Response · Author response to Decision Letter 0]

23 Nov 2023

Reviewers’ comments are numbered and in normal font. Responses are in blue, extracts from the original manuscript are in italic and black while the revised portions are in red.

Reviewer #1:

This is a very interesting but under-researched area of study, and these results have potential clinical implications, so it is welcome that the researchers have decided to conduct this study. Below I describe my main concerns and suggestions.

We thank the reviewer for highlighting that our study is very interesting and has potential impact for clinical applications. We have now amended the manuscript and added discussion sections to provide more supporting evidence for our main conclusions. 

1) Causal inference

In the section “Subjective Causality of Meditation Method Use and PLEs” you talk about a criterion for causality citing WHO 2013, “an event that happens concurrently or right after an intervention.” This addresses one of many causal attribution criteria for assessing adverse effects of pharmacological interventions, but it is not really sufficient on its own – please reflect this in your wording. The Varieties of Contemplative Experience paper discusses causal attribution criteria in greater detail (see https://doi.org/10.1371/journal.pone.0176239 page 28), and subjective judgement of causality is just one of eleven factors used in that study to determine causality.

In discussing causality for this study it is overstretching to talk about a causal relation between meditation method and PLEs. It would be good to consistently use language like “perceived causality” should be used to discuss the finding that participants felt that a method of meditation was causing the experience of a PLE. Additionally, these questions should not be asked of participants who have not had a PLE, because their inclusion in the data skews your results further toward hypothetical perception of a relationship, rather than an empirical one, derived from first-hand experience of those who have had one. You also imply that this relationship is a fact of which participants are subjectively aware in the first paragraph of the discussion. There is not sufficient evidence for this claim.

We thank the reviewer for highlighting the potential issues with our causality assessment, which we initially tried to address using the term “subjective causality” but realise now this might not have been sufficient. We have carefully edited our whole manuscript using the term “perceived causality” and added a limitation section in our results, which we have pasted under the 4th comment of the reviewer. 

In order to test the potential issue of including participants who never experienced PLEs in our subjective causality analysis, we also repeated our analysis using only participants who experienced at least 25% of positive symptoms of PLEs (mean CAPE score > 29). Our results did not change. We are now presenting this new analysis in the paper. 

Perceived Causality of Meditation Method Use and PLEs

The analyses above establish correlations between the types of meditation and PLEs, but we also wanted to test perceived causality (hypothesis 1c). There are different ways to assess causality (WHO 2013), here we focus on a specific aspect, namely subjective attribution. To assess perceived causality, we asked participants to rate their perception of causality between the different meditation methods and PLEs (Fig. 2D) using a scale running from -5 to 5 (‘Have you found your meditation practice to either make these experiences less likely or more likely to happen?’).

We found, indeed, that meditation techniques differed in their perceived causality. First, we related the meditation categories (NDM, CDM, ADM) to subjective ratings of how ‘preventative’ of PLEs the use of techniques within each category were. Doing so, we found that a model with the scores of how preventative each meditation category was differed significantly from one without the categories (F(2) = 18.58, p < .001, S3).

Post hoc analysis revealed that all three meditation methods were rated as preventative of PLEs in daily life (S6). However, our results also show that NDM was perceived as the least preventative method compared to the other two (Fig 2D, S6 and 7, NDM vs CDM: t836 = -5.48 p < .001; NDM vs ADM: t836 = -5.04 p < .001). As a control analysis, to test the perceived causality between meditation categories and PLEs only for those individuals who had experienced PLEs, we repeated the same analysis by only including meditators with at least 25% of PLEs scored as positive (n = 391; Mean CAPE positive score > 29). Our results showed that the perceived causality remained the same; the difference between NDM and CDM was significant (t1818 = -5.33, p < .001), as well as NDM and ADM (t1818 = -4.86, p < .001) (S8). All meditation methods also remained significantly different from zero (S9). 

We have also added a new section in our Figure 1C to illustrate the distribution of PLEs:

Figure 1. A. Meditation methods according to Nash, Newberg and Awasthi’s taxonomy27 (NDM= Null Directed Methods, CDM = Cognitive Directed Methods, ADM = Affective Directed Methods). B. Total lifetime use for each meditation method across participants from ratings 0 (Not at all) to 10 (Very often) for each of the five in each category and summed (i.e. minimum is 0 and maximum is 50). A participant could score highly in only one category or in several or in none. C. Distribution of PLE (CAPE positive) scores with a cut-off (red dotted line) for clinical significance . D. Pre-registered analyses: from questionnaires, psychotic-like experiences (PLEs)15, and mystical experiences37, were extracted to test how they were differentially affected by different types of meditation. E. Spread of overall lifetime meditation practice in years, with the dotted vertical line marking 10+ years experience.

We also added this new analysis to the discussion section:

Discussion

In a pre-registered large online survey (n=613), we tested whether distinct types of meditation methods differentially correlated with PLEs and mystical experiences. We first applied a previous taxonomy based on the goal states of meditation methods27 and grouped meditation methods26 into Null Directed Methods (NDM), Cognitive Directed Methods (CDM) and Affective Directed Methods (ADM). We found that lifetime use of NDM methods, which aim at achieving an altered state of consciousness devoid of phenomenological content27, correlated with PLEs as measured by the CAPE-42 positive symptom scale. Lifetime use of CDM methods showed a negative and statistically significant relationship with PLEs in daily life. We also found that participants were subjectively aware that the different meditation methods have different relationships to adverse experiences, that we term ‘perceived causality’. Indeed, they rated CDM and ADM methods preventative of PLEs, and while NDM was also rated as preventative overall, it was rated as significantly less so than CDM or ADM. This held true for the whole sample (n = 613), as well as for a sample including only those scoring 25% or above on the mean PLE score. Taken together, these findings suggest that the use of meditation methods with a focus on attaining an ‘empty state’ devoid of phenomenological content correlates with an increase in PLEs, while methods aimed at an enhanced cognitive state showed the opposite correlation.

2) Operationalise Your Terms

a) It would be good to clearly define, for the sake of the study, what definitions you’re using for mystical experiences and PLEs. You do explore the idea that PLEs are the positive aspects of psychosis in a parenthetical statement in your intro, but less discussion is given to what a mystical experience is—perhaps the working definition of both terms is the same and you’re curious how survey participants define them differently. Add a paragraph or two explicitly operationalising your key terms, what working definition you used, and how you measured it (in addition to your later section that describes each of the scales). This is done somewhat in the section called “Differences Between Mystical Experiences in Subjects Distress and Positive Ratings,” but not sufficiently. Are there positive and negative mystical experiences, or have you defined a mystical experience as a positive PLE?

We have now added a new section in the methods to explain our terms more clearly, as:

Mystical Experiences Scale

Mystical experiences have been associated with meditation35, so the Mystical Experiences Scale37 was used to capture these experiences, such as feeling connected to all living things, a sense of ineffability and alterations to the experience of time and space. The 30-item revised Mystical Experience Questionnaire (MEQ30) used in this study consists of four subscales and was derived from a forty-three-item version37. The MEQ has four subscales including ‘Mystical’, 'Positive Mood’, 'Transcendence of Time and Space’ and ‘Ineffability’. The Mystical subscale (which has items from the internal unity, external unity, noetic quality, and sacredness scales of the MEQ43), relates to the suggestion that ‘mystical’ experiences reported by both religious and non-religious individuals in the literature (e.g. Stace’s 1960 framework) can be characterised predominantly by a deep sense of ‘interconnectedness’ and ‘unity’75. This interconnectedness relates both to an internal and external sense of a ‘oneness’ representing ‘God’ or the ‘Universal Self’. The noetic quality aspect relates to a sense that the experience comes from a source of objective truth, noetic meaning of or relating to the intellect. The sacredness aspect relates to the sense the experience is worthy of reverence75. The Positive Mood subscale relates to reports of a deep sense of peace and joy accompanying mystical experiences, and the Transcendence of Time and Space subscale relates to experiences of the perception of the boundaries of time and space altering or becoming limitless. The Ineffability subscale relates to the reported quality of mystical experiences being difficult or impossible to put into words or explain to others75.

Scoring runs from 0-None; not at all to 5-Extreme (more than any other time in my life and stronger than 4), with a highest total of one hundred and fifty with higher scores indicating more self-reported mystical experiences. This measure was designed to assess such experiences whilst using psychedelic substances, which can be seen by the wording of the rating scale. We adapted the instructions at the start of the scale so that it referred to daily experiences, but left the rating scale as it was:

- “Please read each statement and rate how much this has ever applied to you in daily life, and then during/immediately after meditation. The rating scale is as follows:”

The original instructions:

- “Looking back on the entirety of your session, please rate the degree to which at any time during that session you experienced the following phenomena. Answer each question according to your feelings, thoughts, and experiences at the time of the session. In making each of your ratings, use the following scale:”

Introduction addition outlining a clearer definition of PLEs and how we measured them:

Psychotic-like experiences (PLEs) are classified as symptoms that are at a subthreshold for diagnosis for a psychotic episode, with increased prevalence of such experiences signifying an increased risk of psychosis11. PLEs are considered part of a continuum of psychosis, and people reporting such experiences do not always report distress or dysfunction12, indeed it is estimated that 26.69% of the general population have had at least one PLE13 in their lifetime. Yet, PLEs are also a risk factor for transitioning to a psychotic disorder14. PLEs can be measured by screening questionnaires such as the CAPE-4215 . They consist of positive symptoms presented as hallucinations (visual and auditory), delusions and thought disorder, and negative symptoms including anhedonia, withdrawal from social interaction and depressive symptoms such as rumination16. The CAPE-42 has three subscales which are ‘Positive’, ‘Negative’ and ‘Depressive’17. We focus here exclusively on the positive subscale, i.e. symptoms related to hallucinations and delusions. Those at risk of psychosis exhibit cognitive biases that are found in people with psychosis. These cognitive biases include aberrant salience, a misattribution of meaning to irrelevant stimuli, heightened attention to threat, and the externalising bias which leads people to misattribute internally generated stimulus to external sources16. A genetic predisposition to psychosis has been reported18, but environmental factors including childhood trauma and immigration status have also been associated with increased risk of psychosis spectrum disorders19. Poorer sleep quality has been linked to symptom onset and severity in early psychosis20, highlighting how environmental factors play a role for those potentially at risk.

b) There is not sufficient treatment of the PLEs at the same time being desirable and a source of insight on the one hand, and on the other, constituting a risk against which certain methods of meditation constitute a protective factor. It would be appropriate to address why some people are seeking PLEs/mystical experiences, and some people may wish to avoid them.

This is a good point regarding why some might see PLEs and Mystical experiences as less distressing/desirable. Our data collection helped answer this question, specifically the ‘reasons to start meditating’ data is relevant here. We have taken two approaches to address this comment, one is to expand on a discussion about the overlap between mystical experiences and PLEs referring to previous literature, and the other is to present data related to subjective distress Vs positive ratings of PLEs and how this links to reasons people started meditating. For the first approach, we will include a discussion related to previous literature in the introduction and discussion sections as outlined below. The reasons to meditate and distress/positive ratings we have included in the results section, and expanded on our factor analysis we ran on the reasons to meditate in the methods section and results. Both are then discussed in the discussion section as highlighted below. We have also addressed a later question about data from other questionnaires not yet included in this article under comment 5) Statistical Analysis.

Introduction addition discussing PLEs, distress and insights, and individual differences:

In a community sample of teenagers reporting at least one psychotic-like experience, 75% found that experience distressing38. Distress from a psychotic episode is reported to stem from both intrapersonal and interpersonal sources39. Intrapersonal sources include unwanted internal states, alterations to identity and self, and disruption to goals and behaviours. Interpersonal states included impacts upon relationships, and stigma39. Persecutory delusions have been found to be a significant predictor of high anxiety and worry40. Indeed, it is reported that heightened anxiety precedes the formation of delusional thinking41. Such evidence suggests that PLEs are very likely to cause distress and link to anxiety in both the clinical40 and non-clinical populations38, although as we highlighted this might link to ‘appraisals’ of such symptoms and how prevalent paranoid ideation is for the individual42.

Psychosis has also been reported to lead to ‘personally transformative growth’ for some people43. This may happen by taking a purposeful approach to learning from such experiences, and with a ‘mindful’ perspective43. Delusions have been suggested to hold personal meaning and help make sense of people's unusual experiences, rather than being nonsensical confabulations, which challenges the traditional view44. Religious and spiritual beliefs have been shown to be important aspects of understanding a person's own experience of psychosis, and sometimes alongside a medical diagnosis45. This demonstrates perceived distress related to PLEs might vary, and may be linked to attitudes towards such experiences43 re

---

## [Decision Letter · Decision Letter 1]

23 Jan 2024

PONE-D-23-11057R1Differential Relationship Between Meditation Methods and Psychotic-Like and Mystical ExperiencesPLOS ONE

Dear Dr. Palmer,

Thank you for submitting your manuscript to PLOS ONE. After careful consideration, we feel that it has merit but does not fully meet PLOS ONE’s publication criteria as it currently stands. As you can see from the comments of both reviewers, there are major methodological and theoretical issues that need to be addressed. Therefore, we invite you to submit a revised version of the manuscript that addresses the points raised during the review process.

We look forward to receiving your revised manuscript.

Kind regards,

Rakesh Karmacharya, MD, PhD

Academic Editor

PLOS ONE

Reviewers' comments:

Reviewer's Responses to Questions

**Comments to the Author**

1. If the authors have adequately addressed your comments raised in a previous round of review and you feel that this manuscript is now acceptable for publication, you may indicate that here to bypass the “Comments to the Author” section, enter your conflict of interest statement in the “Confidential to Editor” section, and submit your "Accept" recommendation.

Reviewer #1: (No Response)

Reviewer #2: (No Response)

2. Is the manuscript technically sound, and do the data support the conclusions?

Reviewer #1: Partly

Reviewer #2: Yes

3. Has the statistical analysis been performed appropriately and rigorously? 

Reviewer #1: I Don't Know

Reviewer #2: Yes

4. Have the authors made all data underlying the findings in their manuscript fully available?

Reviewer #1: Yes

Reviewer #2: Yes

5. Is the manuscript presented in an intelligible fashion and written in standard English?

Reviewer #1: Yes

Reviewer #2: No

6. Review Comments to the Author

Reviewer #1:

 Thanks again for inviting me to comment on this interesting work. This new version represents significant progress. The authors have worked hard adding important information in a number of places. I am afraid, however, that the manuscript needs further work, as detailed below.

• The paper is very hard to parse, it needs more attention in order to avoid confusion. For example, some methods are explained in the introduction and in the results. Also, results are presented without introducing them in methods: Table 2 presents pearson correlations not introduced in the methods or linked with anything else.

• Plos One papers have the methods before the results, please move them.

• Regression weights and coefficients are not the same, please use word coefficients consistently.

• Please clarify in the methods that the t tests are on regression coefficients. As far as I understand, what is usually reported in regressions is the standardised beta coefficient and the p value, not the t value. And why are effect sizes calculated using partial eta squared and not simply interpreting the regression coefficients? I suggest you report the beta coefficients instead of the t values, as they are more interpretable. it is good practice to report 95% confidence intervals too.

• Have the authors tested whether regression models were meeting statistical assumptions (eg distribution of residuals, etc)? There is no mention of this important aspect that can invalidate results if assumptions are not met.

• Reasons for meditating seem to have been classified into two factors, but there is no explanation of what these factors are (only variable names in the graphs).

• Regarding making causal claims about meditation type and PLEs: The section on hypothesis 1c seems clear with regards to perceived causality, but in the Discussion (on page 3) authors argue they have evidence for causality, " We also found that participants were subjectively aware that the different meditation methods have different relationships to adverse experiences, that we term ‘perceived causality’." this should rather be stated, "Although cross-sectional survey data can't prove a causal relationship between meditation type and PLEs, we have found that meditators perceive that different types of meditation may be more likely to elicit PLEs than others."

• They have a much better discussion of the potential connections and differences between mystical experiences and PLEs.

• I'd encourage them to use one word like "perceive" just in context of subjective causal attribution and different word like "appraise" in terms of subjective valence (pos/neg), otherwise. For instance this sentence on the top of page 10, "For mystical experiences, people who experienced more PLEs perceived mystical experiences as more distressing (t138 = 2.95, p <.01, S20), with no difference (or even an opposite trend) to how positively they were perceived."

• The limitations section needs much more thought. Important ones that are not mentioned, are that the cross-sectional nature does not allow to understand the temporal sequence of factors, and that these relationships can all be confounded by unmeasured factors. Both have important implications that need to be reflected upon and discussed.

Reviewer #2: 

The authors have extensively reworked this manuscript, and these changes have led to a significant improvement. However, a number of problems remain which I would like to see addressed before recommending this manuscript for publication. 

General comments:While some of the additional information provided in this version of the manuscript is important, some of the sections have become quite wordy and excessively long. For example, the scale descriptions in the methods section are an important addition, but need to be further edited to be more concise and parsimonious. Similarly, the analysis section of the methods feels excessively long and in need of further editing. I would like to see this section written in full sentences and paragraph form. The manuscript could benefit from additional proofreading, as there are awkward sentences, typos, and grammatical errors throughout. The header structure is at times unclear and confusing. For example, in the methods section, it appears (but I’m not sure) that “Questionnaires Included” is a separate section after “Materials and Measures.” I would recommend using a more standard header structure and providing some additional subheaders in the introduction and discussion. For example, the transition from the second to third paragraph of the introduction feels very abrupt and could either benefit from a header or further editing. The new sections of the introduction on the valence of psychotic or mystical experiences also seem unnecessarily long for this point in the introduction (see additional comments below about this novel section). As a general point, I think the introduction needs additional framing to contextualize some of the results that are not introduced at all in the introduction, while this novel section is important but currently overrepresented in the introduction. Theoretical Issues:There are theoretical problems in the novel section of the introduction about the valence of psychotic and mystical experiences. Unfortunately, the authors seem to have misunderstood my previous point about the overlap between mystical and psychotic-like experiences. Page 3 of the introduction discusses the incidence of distress vs. “personally transformative growth” as a result of psychosis. However, this conflates the general construct of “psychosis” with the specific CAPE-42 items about hallucinations and delusions that this study uses as its measure of PLEs. “Psychosis” is already a pathologizing term, and as a clinical diagnostic category, it is defined in terms of functional impairment or distress. Instead, I would like to know whether the CAPE-42 items used in the current study are worded in a way that assumes/implies they are negative/pathological or whether they are phenomenologically neutral. This is important, as visions and magical thinking (another way of phrasing hallucinations and delusions) are valued in many spiritual traditions. If the items are neutrally worded, it is therefore unclear whether they are measuring something innately pathological or not. The next paragraph about mystical experience in the introduction has the same theoretical problems. Given that this study uses the MEQ30 to measure mystical experiences, mystical experiences should be understood as whatever is being measured by these items (which notably do not include the word God). Therefore, I would like to see the discussion of mystical experiences less tied to an abstract notion and more related to what is specifically being measured in this study. More importantly, the MEQ includes positive affect in its definition, which makes a discussion of whether mystical experiences are positive for some people or distressing for others nonsensical when based on this measure and definition, given that the construct is *defined* as being innately positive. I would encourage a critique of the way that mystical experiences have been defined, perhaps in the discussion or limitations, but given that this definition is already being used, it does not make theoretical sense to discuss its valence. Finally, please replace the Parnas & Hendriksen citation with a more up-to-date citation on the sense-of—self-related similarities between mystical and pathological experiences or leave out this citation entirely, as current scholarship on the topic has advanced considerably since this was published. Here is what I would recommend in response to these issues: introduce the MEQ and CAPE-42 as being innately biased toward positive or negatively valenced experience (if that is also the case for the CAPE-42) and comment on how they share some common phenomenology (if they do, at the item level). Then, throughout the rest of the paper, frame PLEs and mystical experience as related but each biased toward pathological/negative appraisals or spiritual/positive appraisals.  Finally, the “exploratory analysis 2: differences between mystical experiences in subjects distress and positive ratings” should be removed, as it is redundant to ask about the valence of mystical experiences when the measure already includes positive affect. The section on the discussion that covers these topics also needs to be edited based on the issues identified above. Please discuss what is actually being measured in this study rather than assuming that it refers to broader abstract concepts. For example:

i.The sentence “It could be argued that mystical experiences are not necessarily positive or negative and in many cases might be neutral” doesn’t make sense when mystical experiences are *defined* as involving positive affect, though authors have pointed out that this definition is problematic (see Taves, 2020).  

ii.“PLEs and mystical experiences overlap in terms of their link to altered experiences of self” – please specify if this is the case, and if so which experiences of self, for the two specific measures used in this study. 

iii.Likewise, please link the discussion of interoception more closely to the specific items/subscales used to measure interoception. For example, feeling safe in one’s body might not be the same thing as the felt sense of presence in one’s body (see self review article: Britton et al., 2021). There are many forms of interoception and it is problematic to assume that the measure that this study uses for interoception corresponds to theoretically different conceptualizations of interoception. 

Issues with framing the results:The hypothesis section of the introduction feels quite lengthy and many of the hypotheses were not framed or supported in the introduction until that point. I am confused by why there is a paragraph describing hypotheses and then a long list of many of the same hypotheses below. Maybe the longer list could be moved to supplemental materials. I also think 1a, 1b, and 1c could be condensed into a single paragraph perhaps. I am confused about why the methods section is below the results and discussion. I think it would be easier to follow the results if I understood the methods first, especially the measures section (with example items to illustrate exactly what is being measured). In the results section, I think it would read better if much of the information in the table/figure captions was moved to the body text. The table/figure captions are very long and hard to follow and the paragraphs do little to introduce the tables/figures. I think the paper would benefit from more introduction and framing of the exploratory results section. I would like to see some framing in the introduction that prepares the reader for these results, introduces the concepts of interoceptive awareness, decentering, and reasons for meditating and why they might be important for the research questions. The limitations section is quite short and could include quite a lot more. I’d like to see more limitations about the sample (how accurately does it generalize to a larger population), the analyses used, the measures used, and the theory that distinguishes mystical experiences from psychosis. Finally, I’d like to see a conclusion section after the limitations.**********

---

## [Author Response · Author response to Decision Letter 1]

22 May 2024

Reviewer #1:

 Thanks again for inviting me to comment on this interesting work. This new version represents significant progress. The authors have worked hard adding important information in a number of places. I am afraid, however, that the manuscript needs further work, as detailed below.

• The paper is very hard to parse, it needs more attention in order to avoid confusion. For example, some methods are explained in the introduction and in the results. Also, results are presented without introducing them in methods: Table 2 presents pearson correlations not introduced in the methods or linked with anything else.

We agreed with the reviewer that the paper was very dense after the revisions. We have tried to clarify the text as much as possible to avoid duplicates. We have also created a supplementary methods section where Table 2 Pearson’s correlation are explained:

Meditation Taxonomy Corelations

We ran a Pearson correlation coefficient for all the meditation techniques with PLEs. We then took the top 5 positively correlating techniques and the top 5 negatively correlating techniques and categorised them according to 3 taxonomies. We applied Nash et al’s23, Dahl et al’s24 and Matko’s25 taxonomies to these techniques. We then took the techniques within the categories we defined in our study, and which we used for our main analysis for hypothesis 1a and applied the same approach of categorising according to Mako and Dahl’s taxonomies. The aim was to explore whether there were any patterns in how techniques with either positive or negative correlations to PLEs were categorised according to different taxonomies.

• Plos One papers have the methods before the results, please move them.

We have now done so.

• Regression weights and coefficients are not the same, please use word coefficients consistently.

We agree it is best to use a consistent term throughout and have replaced weights with coefficients throughout.

• Please clarify in the methods that the t tests are on regression coefficients. As far as I understand, what is usually reported in regressions is the standardised beta coefficient and the p value, not the t value. And why are effect sizes calculated using partial eta squared and not simply interpreting the regression coefficients? I suggest you report the beta coefficients instead of the t values, as they are more interpretable. it is good practice to report 95% confidence intervals too.

We chose eta-squared over standardized beta coefficients as this has a more intuitive interpretation and there are standard tables to characterize effect sizes as ‘small’, ‘medium’ or ‘large’. Standardized beta coefficients (i.e. the regression coefficients obtained when regressors are standardized [i.e. subtracting the mean, dividing by standard deviation]) can range from minus to plus infinity, thus are less intuitive. We include beta coefficients in the regression output tables in the sup.

• Have the authors tested whether regression models were meeting statistical assumptions (eg distribution of residuals, etc)? There is no mention of this important aspect that can invalidate results if assumptions are not met.

We have now addressed this concern with a section in the supplements, and a results table of a new analysis controlling for heteroscedasticity as S24 in the supplements:

Further Control Analyses - regressions

For each model, we checked the assumptions of linear regression, i.e., linearity (visual inspection of scatter plot of variables of interest), multicollinearity (inspection of correlation tables of regressors, checking for abs(r)>0.5), homoscedasticity (Breusch-Pagan Test) and normality of residuals (Shapiro-Wilk test). We found that all the pre-registered hypotheses violated assumptions of normality and homoscedasticity. As mentioned in the main text, there were no issues with multicollinearity.

According to Li and Ding42, the Central Limit Theorem provides that, with sufficiently large sample sizes, the sampling distribution of the mean will approximate a normal distribution, regardless of the shape of the underlying distribution. Previous work has highlighted the robustness of linear regression to violations of normality (other than for small samples, e.g., n<10)43,44. Despite this fact, we decided to add a control test to account for violations of homoscedasticity. To compute the significance of the regression coefficients we used heteroscedasticity-consistent (HC) standard errors implemented in the sandwich package in R45. We find that all regression coefficients remain significant in our pre-registered hypotheses with one exception for hypothesis 3 (psychosis proneness measures) with the dissociative absorption (DAS) score. This changed from a significance of p <.05 to .06 (S24).

• Reasons for meditating seem to have been classified into two factors, but there is no explanation of what these factors are (only variable names in the graphs).

We apologise for the lack of clarity. This was in the methods, on section titled “Factor Analysis for Reasons to Start Meditating”(page 6). We hope that with the restructuring (methods first) this is now easier to find. There is a table in the supplements with these reasons placed into relevant factors, that we have referred to in the text (S3). 

• Regarding making causal claims about meditation type and PLEs: The section on hypothesis 1c seems clear with regards to perceived causality, but in the Discussion (on page 3) authors argue they have evidence for causality, "We also found that participants were subjectively aware that the different meditation methods have different relationships to adverse experiences, that we term ‘perceived causality’." this should rather be stated, "Although cross-sectional survey data can't prove a causal relationship between meditation type and PLEs, we have found that meditators perceive that different types of meditation may be more likely to elicit PLEs than others."

This has been changed as suggested.

• They have a much better discussion of the potential connections and differences between mystical experiences and PLEs.

We thank the reviewer for their positive comment.

• I'd encourage them to use one word like "perceive" just in context of subjective causal attribution and different word like "appraise" in terms of subjective valence (pos/neg), otherwise. For instance this sentence on the top of page 10, "For mystical experiences, people who experienced more PLEs perceived mystical experiences as more distressing (t138 = 2.95, p <.01, S20), with no difference (or even an opposite trend) to how positively they were perceived."

This has been changed as suggested. The references to distress and PLEs Vs mystical have been removed in relation to a comment by the other reviewer.

• The limitations section needs much more thought. Important ones that are not mentioned, are that the cross-sectional nature does not allow to understand the temporal sequence of factors, and that these relationships can all be confounded by unmeasured factors. Both have important implications that need to be reflected upon and discussed.

We have added two important sections in the limitations section as suggested (Page 22).

Reviewer #2: 

The authors have extensively reworked this manuscript, and these changes have led to a significant improvement. However, a number of problems remain which I would like to see addressed before recommending this manuscript for publication. 

1. General comments:

a. While some of the additional information provided in this version of the manuscript is important, some of the sections have become quite wordy and excessively long. For example, the scale descriptions in the methods section are an important addition, but need to be further edited to be more concise and parsimonious. Similarly, the analysis section of the methods feels excessively long and in need of further editing. I would like to see this section written in full sentences and paragraph form. 

We have a more succinct description of measures in the main article (Page 6), with a fuller more detailed description in the supplementary methods section (Questionnaires Included). We have edited the analysis section of the main article (Page 8 up to and including Page 11) and included a more detailed version in the supplementary methods section (Statistical analysis and deviations from pre-registration). The section in the main article is now cleared titled and more succinct, written in full sentences.

Here is an example of how we have now formatted it in the main text:

Meditation Method and PLEs (hypothesis 1a)

We used multiple linear regression to investigate the relationship between types of meditation (independent variables) and the main daily-life PLE measure (positive scale from CAPE-42 and 3 items about visual hallucination from LSHS36). The dependent variable was the sum score of the PLE measure (total score per participant on the CAPE). The independent variables were meditation categories defined according to Nash et al22 i.e. 3 categories . Pre-registered hypothesis 1b, i.e. the same test for PLEs during meditation could not be tested due to an error in phrasing the question (see methods section in supplements).

PLEs ~ NDM Meditation Use + CDM Meditation Use + ADM Meditation Use + Age + Sex

b. The manuscript could benefit from additional proofreading, as there are awkward sentences, typos, and grammatical errors throughout. 

Thank you for your careful read, we have now reviewed our entire paper for typos and grammatical issues.

c. The header structure is at times unclear and confusing. For example, in the methods section, it appears (but I’m not sure) that “Questionnaires Included” is a separate section after “Materials and Measures.” I would recommend using a more standard header structure and providing some additional subheaders in the introduction and discussion. For example, the transition from the second to third paragraph of the introduction feels very abrupt and could either benefit from a header or further editing. 

We have now substantially rewritten the introduction (Page 3 up to and including Page 4) which is now shorter and edited the methods section (Page 5 up to and including Page 11). We believe that it reads a lot better now.

d. The new sections of the introduction on the valence of psychotic or mystical experiences also seem unnecessarily long for this point in the introduction (see additional comments below about this novel section). As a general point, I think the introduction needs additional framing to contextualize some of the results that are not introduced at all in the introduction, while this novel section is important but currently overrepresented in the introduction. 

We agree with the reviewer. Please see comment above as this is relevant for this new one. We have removed a lot of this discussion relating to valence of PLEs and mystical experiences. We also stated how we are specifically measuring these experiences:

Here we measure mystical experiences using the Mystical Experiences Questionnaire (MEQ) and psychotic-like experiences with the Community Assessment of Psychic Experiences-42 (CAPE) positive subscale with three extra items from the Launay-Slade Hallucination Index (LSHI)36.

2. Theoretical Issues:

a. There are theoretical problems in the novel section of the introduction about the valence of psychotic and mystical experiences. Unfortunately, the authors seem to have misunderstood my previous point about the overlap between mystical and psychotic-like experiences. Page 3 of the introduction discusses the incidence of distress vs. “personally transformative growth” as a result of psychosis. However, this conflates the general construct of “psychosis” with the specific CAPE-42 items about hallucinations and delusions that this study uses as its measure of PLEs. “Psychosis” is already a pathologizing term, and as a clinical diagnostic category, it is defined in terms of functional impairment or distress. Instead, I would like to know whether the CAPE-42 items used in the current study are worded in a way that assumes/implies they are negative/pathological or whether they are phenomenologically neutral. This is important, as visions and magical thinking (another way of phrasing hallucinations and delusions) are valued in many spiritual traditions. If the items are neutrally worded, it is therefore unclear whether they are measuring something innately pathological or not. 

We agree this is a complex issue, and it may be that people may frame or interpret the same type of experience in different ways, and that levels of distress associated with such experience may vary too. We have tried to discuss this point, including a new analysis looking at the wording of the items in the CAPE-42+ and the MEQ and the correlation of these items. 

In short, we believe that even though there can be more “negatively” phrased items in the CAPE-42 and more “positively” phrased items in the MEQ, there remains an item-by-item relationship between the two questionnaires which indicate that generally, regardless of the valence of the items used, people that are more likely to experience PLEs and also the ones that are more likely to experience mystical experiences, which was our original point. We, however, take on board the suggestion and included the additional analyses in results and discussion. We have also streamlined our introduction and removed reference to personally transformative growth.

Another point is that we think the distinction between PLEs and psychosis is important, and have purposefully not used the word psychosis but only PLEs. It is true that PLEs are often reported as distressing, and we thank the reviewer for raising this important point regarding ‘pathology’. We previously included discussion on this point and have now attempted to make this much clearer. This also ties in with a discussion we have also included about negative versus positive biases in terms of specific questionnaires (CAEP-42 and MEQ), and our new analysis.

We have included a new analysis an outlined this in the methods section:

Exploratory analysis 3: Correlations of PLE and mystical measure items (CAPE-42 +3 LSH and the MEQ), and theoretical overlap 

We used all the CAPE-42+ items and the MEQ items to create a correlation matrix (Pearson’s r) to explore the links between individual items of our PLE mystical experiences measures. 

Please see results section titled (Page 19 and 20): 

Exploratory analysis 3: Correlations of Questionnaire Items from the CAPE-42+ and the MEQ

Our discussion section now includes a discussion of our new analysis in the 4th paragraph.

b. The next paragraph about mystical experience in the introduction has the same theoretical problems. Given that this study uses the MEQ30 to measure mystical experiences, mystical experiences should be understood as whatever is being measured by these items (which notably do not include the word God). Therefore, I would like to see the discussion of mystical experiences less tied to an abstract notion and more related to what is specifically being measured in this study. More importantly, the MEQ includes positive affect in its definition, which makes a discussion of whether mystical experiences are positive for some people or distressing for others nonsensical when based on this measure and definition, given that the construct is defined as being innately positive. I would encourage a critique of the way that mystical experiences have been defined, perhaps in the discussion or limitations, but given that this definition is already being used, it does not make theoretical sense to discuss its valence. Finally, please replace the Parnas & Hendriksen citation with a more up-to-date citation on the sense-of—self-related similarities between mystical and pathological experiences or leave out this citation entirely, as current scholarship on the topic has advanced considerably since this was published. 

We have included the aforementioned new analysis of the item-by-item relationship of the MEQ and CAPE as discussed. This includes discussion of the valence of each questionnaire. The analysis of distress ratings of MEQ and CAPE has been removed. The citation mentio

---

## [Decision Letter · Decision Letter 2]

23 Jun 2024

PONE-D-23-11057R2Differential Relationship Between Meditation Methods and Psychotic-Like and Mystical ExperiencesPLOS ONE

Dear Dr. Palmer,

Thank you for submitting your manuscript to PLOS ONE. After careful consideration, we feel that it has merit but does not fully meet PLOS ONE’s publication criteria as it currently stands. Therefore, we invite you to submit a revised version of the manuscript that addresses the points raised during the review process. As both reviewers note, the quality of the manuscript has improved significantly. Your revised manuscript has addressed all the issues raised by Reviewer 1 as well as many of the issues raised by Reviewer 2. There are a few important issues that remain in relation of the concerns raised by Reviewer 2, as described below, many of which are related primarily to the style and presentation of the content in the manuscript and a few missing details.  

We look forward to receiving your revised manuscript.

Kind regards,

Rakesh Karmacharya, MD, PhD

Academic Editor

PLOS ONE

Reviewers' comments:

Reviewer's Responses to Questions

**Comments to the Author**

1. If the authors have adequately addressed your comments raised in a previous round of review and you feel that this manuscript is now acceptable for publication, you may indicate that here to bypass the “Comments to the Author” section, enter your conflict of interest statement in the “Confidential to Editor” section, and submit your "Accept" recommendation.

Reviewer #1: All comments have been addressed

Reviewer #2: (No Response)

2. Is the manuscript technically sound, and do the data support the conclusions?

Reviewer #1: (No Response)

Reviewer #2: Yes

3. Has the statistical analysis been performed appropriately and rigorously? 

Reviewer #1: (No Response)

Reviewer #2: Yes

4. Have the authors made all data underlying the findings in their manuscript fully available?

Reviewer #1: (No Response)

Reviewer #2: Yes

5. Is the manuscript presented in an intelligible fashion and written in standard English?

Reviewer #1: (No Response)

Reviewer #2: No

6. Review Comments to the Author

Reviewer #1: All my comments have been addressed, thank you. I look forward to seeing this manuscript published as it will be an interesting contribution to the field.

Reviewer #2: Thank you for the opportunity to review this interesting manuscript. This revision represents a significant improvement over the previous draft. The authors have clearly put a lot of work into responding to the comments from the previous round of revisions. Many of the technical and theoretical issues have now been sufficiently addressed in my opinion. Furthermore, the results of this study are interesting, important, and novel. I believe that the study does merit publication. Unfortunately, however, I still think it needs significant work in the following areas before it is ready for publication:

Writing quality: the manuscript needs a great deal of further proofreading and formatting.

1. There are still grammatical issues, run-on sentences, and spelling mistakes throughout the manuscript. For example, decentering is spelled incorrectly on second page of introduction, “and” is spelled incorrectly in the title of Table 2, capitalization is inconsistent, etc.

2. The manuscript needs additional formatting according to APA style or other journal standards. For example, sentences should be in past tense, tables and figures should be referenced and contextualized in the text, and headers should be structured in a more standard way with main headers and subheaders

3. Acronyms should be introduced first with the full name (e.g., CAPE-42 in the introduction).

4. The language in many places needs to be clearer and more precise. For example, please clarify differences between “related to, linked to, defined as, correlated with” which seem to be used interchangeably at times (especially in second page of introduction). Describing how a construct is defined should be clearly delineated from describing what it is correlated with.

5. The study should not repeat so many times that it is pre-registered. This can be mentioned in the methods once. Also, avoid language like “a very large sample” and instead just state the size of the sample.

Introduction:

1. I feel that the introduction still needs some work as it is too focused on describing the present study rather than introducing the concepts and theory necessary to contextualize the present study. There are a number of sentences in the introduction that would be better placed in the methods or results. For example on page 1, the following sentence describes the study method and should not be in the introduction: “Here, we examined the impact of different types of meditation methods on PLEs. To allow statistical tests, we grouped them according to a taxonomy proposed by Nash, Newberg and Awasthi.” The last sentence in the introduction is about the results and should be moved to that section.

2. The introduction needs to introduce and describe the concepts more clearly. For example, what is an enhanced cognitive state? Why are MBSR and MBCT considered CDM meditation (and how do you know this since these interventions involve multiple techniques)?

3. The first paragraph of the second page about reasons to meditate provides little theory or context and instead focuses on the study’s research questions. Why are reasons for meditating important? What do other studies show about reasons to meditate?

4. Given the importance placed in Table 2 on the three different meditation taxonomies, I think these need to be described in more detail in the introduction in order to provide more context for the rest of the paper.

5. The concepts of decentering and interception are very briefly mentioned and also need to be further defined. Why would these core mechanisms possibly relate to PLEs? Why are these considered core mechanisms?

6. I think the introduction could say something more about the effects of meditation on altered states. The framing of the first paragraph says that the study looks at the links between different types of meditation and adverse and beneficial outcomes. While this is technically correct, it feels a bit odd not to mention that both of these outcomes (PLEs and mystical experiences) are types of altered states. In general, I think the introduction could benefit from a more in-depth literature review, which would also improve the quality of the discussion.

Methods:

1. Analyses section of methods should be under a single header with subheaders for the various types of analyses.

2. Questionnaires should be a subheader under materials and methods. Techniques, traditions, reasons for meditating, and each individual questionnaire don’t need separate headers.

3. Questionnaire descriptions are awkwardly written and formatted

4. Please include a bit more about the recruitment strategy. How was the study advertised? This has an impact on the sample characteristics and should also be referenced in the limitations (about how the sample might not generalize to all people).

Results:

1. Demographics: what about race, ethnicity, nationality, first-language, sexual orientation?

2. I still feel that the tables and figures need to be better referenced in the text and more clearly described. Table 2 contains useful information but it presents so many different categorical systems that it took me some time to make sense of. I think this table in particular needs to be better described in the text.

Discussion:

1. The discussion doesn’t mention anything about interoception or decentering. I think these should be discussed since they’re part of the results.

2. In general, the discussion could use more integration with the other literature and theoretical considerations instead of just a summary of the results. Why is body focus important? Why is null focus important? Why do you think reasons for meditating are important?

7. PLOS authors have the option to publish the peer review history of their article (what does this mean?). If published, this will include your full peer review and any attached files.

Reviewer #1: No

Reviewer #2: **Yes: **Nicholas K Canby

---

## [Author Response · Author response to Decision Letter 2]

8 Aug 2024

Reviewer #1: All my comments have been addressed, thank you. I look forward to seeing this manuscript published as it will be an interesting contribution to the field.

We thank reviewer 1 for their suggested edits and additions, we believe this has improved the manuscript. We have been careful not to edit or remove any of the previous changes suggested by reviewer 1 in the latest round of edits. Some of the previous changes that we added to the supplements may now be in the main text, as this is one of the biggest changes made.

Reviewer #2: Thank you for the opportunity to review this interesting manuscript. This revision represents a significant improvement over the previous draft. The authors have clearly put a lot of work into responding to the comments from the previous round of revisions. Many of the technical and theoretical issues have now been sufficiently addressed in my opinion. Furthermore, the results of this study are interesting, important, and novel. I believe that the study does merit publication. Unfortunately, however, I still think it needs significant work in the following areas before it is ready for publication:

Writing quality: the manuscript needs a great deal of further proofreading and formatting.

We thank the reviewer for his kind words of encouragement, and acknowledgement of work put into previous edits. We have carefully read each comment and done our best to fully address each one appropriately. Large edits have been applied to the introduction and discussion, with the addition of a lot more discussion of background work and relevant concepts. We believe this has improved the article and given it more depth, so we thank the reviewer for their suggestions. For clarity, we have removed sections from the ‘supplements’ including parts of the supplementary intro, results and methods. This is because much of this is now included in the main text in response to some of the requests below.

More generally we have worked to correct grammar, spelling and punctuation. We have adjusted the method section in line with suggestions. Each comment has a reply to outline what we have done and where to find the alterations which are highlighted in red in the manuscript.

1. There are still grammatical issues, run-on sentences, and spelling mistakes throughout the manuscript. For example, decentering is spelled incorrectly on second page of introduction, “and” is spelled incorrectly in the title of Table 2, capitalization is inconsistent, etc.

We have addressed this.

2. The manuscript needs additional formatting according to APA style or other journal standards. For example, sentences should be in past tense, tables and figures should be referenced and contextualized in the text, and headers should be structured in a more standard way with main headers and subheaders.

We have done this.

3. Acronyms should be introduced first with the full name (e.g., CAPE-42 in the introduction).

We have done this.

4. The language in many places needs to be clearer and more precise. For example, please clarify differences between “related to, linked to, defined as, correlated with” which seem to be used interchangeably at times (especially in second page of introduction). Describing how a construct is defined should be clearly delineated from describing what it is correlated with.

We have reduced the number of terms referencing the same thing. We have edited grammar etc.

5. The study should not repeat so many times that it is pre-registered. This can be mentioned in the methods once. Also, avoid language like “a very large sample” and instead just state the size of the sample.

We have removed extra instances of pre-registered and very large sample is gone.

Introduction:

1. I feel that the introduction still needs some work as it is too focused on describing the present study rather than introducing the concepts and theory necessary to contextualize the present study. There are a number of sentences in the introduction that would be better placed in the methods or results. For example on page 1, the following sentence describes the study method and should not be in the introduction: “Here, we examined the impact of different types of meditation methods on PLEs. To allow statistical tests, we grouped them according to a taxonomy proposed by Nash, Newberg and Awasthi.” The last sentence in the introduction is about the results and should be moved to that section.

We removed suggested sentences. And we expanded on discussion of background work.

2. The introduction needs to introduce and describe the concepts more clearly. For example, what is an enhanced cognitive state? Why are MBSR and MBCT considered CDM meditation (and how do you know this since these interventions involve multiple techniques)?

We have expanded on definitions of MBCT and MBSR and why we consider them CDM. This links to the expanded description of meditation taxonomies and how we used them.

3. The first paragraph of the second page about reasons to meditate provides little theory or context and instead focuses on the study’s research questions. Why are reasons for meditating important? What do other studies show about reasons to meditate?

We have added more about reasons to meditate and their importance.

4. Given the importance placed in Table 2 on the three different meditation taxonomies, I think these need to be described in more detail in the introduction in order to provide more context for the rest of the paper.

A more detailed discussion of these taxonomies has been added.

5. The concepts of decentering and interception are very briefly mentioned and also need to be further defined. Why would these core mechanisms possibly relate to PLEs? Why are these considered core mechanisms?

This has been added.

6. I think the introduction could say something more about the effects of meditation on altered states. The framing of the first paragraph says that the study looks at the links between different types of meditation and adverse and beneficial outcomes. While this is technically correct, it feels a bit odd not to mention that both of these outcomes (PLEs and mystical experiences) are types of altered states. In general, I think the introduction could benefit from a more in-depth literature review, which would also improve the quality of the discussion.

We have made considerable changes to the introduction, including discussion of altered states. 

Methods:

1. Analyses section of methods should be under a single header with subheaders for the various types of analyses.

We have done this.

2. Questionnaires should be a subheader under materials and methods. Techniques, traditions, reasons for meditating, and each individual questionnaire don’t need separate headers.

We have done this.

3. Questionnaire descriptions are awkwardly written and formatted

These have been edited.

4. Please include a bit more about the recruitment strategy. How was the study advertised? This has an impact on the sample characteristics and should also be referenced in the limitations (about how the sample might not generalize to all people).

We have included some more detail in the methods section ‘Procedure’ and added to the limitations section.

Results:

1. Demographics: what about race, ethnicity, nationality, first-language, sexual orientation?

We did not collect this information. But we have added this to the limitations section. 

2. I still feel that the tables and figures need to be better referenced in the text and more clearly described. Table 2 contains useful information but it presents so many different categorical systems that it took me some time to make sense of. I think this table in particular needs to be better described in the text.

This table has now been described in some detail in the main text.

Discussion:

1. The discussion doesn’t mention anything about interoception or decentering. I think these should be discussed since they’re part of the results.

This has been added in some detail under the title ‘Decentering and interoceptive awareness’.

2. In general, the discussion could use more integration with the other literature and theoretical considerations instead of just a summary of the results. Why is body focus important? Why is null focus important? Why do you think reasons for meditating are important?

We have added a discussion of the ‘null’ element of NDM and body focus in the discussion under the section titled ‘NDM techniques and low body focus correlate with PLEs’. We added to the discussion about reasons to meditate in the discussion section: ‘Meditating for spiritual reasons correlates with PLEs’.

We have added further subtitles in the discussion to make it clearer.

---

## [Editor Report · Decision Letter 3]

12 Aug 2024

Differential Relationship Between Meditation Methods and Psychotic-Like and Mystical Experiences

PONE-D-23-11057R3

Dear Dr. Palmer,

We’re pleased to inform you that your manuscript has been judged scientifically suitable for publication and will be formally accepted for publication once it meets all outstanding technical requirements.

Kind regards,

Rakesh Karmacharya, MD, PhD

Academic Editor

PLOS ONE
---

## [Editor Report · Acceptance letter]

27 Aug 2024

PONE-D-23-11057R3 

PLOS ONE

Dear Dr. Palmer, 

I'm pleased to inform you that your manuscript has been deemed suitable for publication in PLOS ONE. Congratulations! Your manuscript is now being handed over to our production team.

Kind regards, 

on behalf of

Professor Rakesh Karmacharya 

Academic Editor

PLOS ONE